# A Regularized Conditional GAN for Posterior Sampling in Image Recovery Problems

**Matthew C. Bendel**
Dept. ECE
The Ohio State University
Columbus, OH 43210
bendel.8@osu.edu

**Rizwan Ahmad**
Dept. BME
The Ohio State University
Columbus, OH 43210
ahmad.46@osu.edu

**Philip Schniter**
Dept. ECE
The Ohio State University
Columbus, OH 43201
schniter.1@osu.edu

## Abstract

In image recovery problems, one seeks to infer an image from distorted, incomplete, and/or noise-corrupted measurements. Such problems arise in magnetic resonance imaging (MRI), computed tomography, deblurring, super-resolution, inpainting, phase retrieval, image-to-image translation, and other applications. Given a training set of signal/measurement pairs, we seek to do more than just produce one good image estimate. Rather, we aim to rapidly and accurately sample from the posterior distribution. To do this, we propose a regularized conditional Wasserstein GAN that generates dozens of high-quality posterior samples per second. Our regularization comprises an $\ell_1$ penalty and an adaptively weighted standard-deviation reward. Using quantitative evaluation metrics like conditional Fréchet inception distance, we demonstrate that our method produces state-of-the-art posterior samples in both multicoil MRI and large-scale inpainting applications. The code for our model can be found here: https://github.com/matt-bendel/rcGAN.

## 1 Introduction

We consider image recovery, where one observes measurements $\boldsymbol{y} = \mathcal{M}(\boldsymbol{x})$ of the true image $\boldsymbol{x}$ that may be masked, distorted, and/or corrupted $\boldsymbol{x}$ with noise, and the goal is to infer $\boldsymbol{x}$ from $\boldsymbol{y}$. This includes linear inverse problems arising in, e.g., deblurring, super-resolution, inpainting, colorization, computed tomography (CT), and magnetic resonance imaging (MRI), where $\boldsymbol{y} = \boldsymbol{A}\boldsymbol{x} + \boldsymbol{w}$ with known linear operator $\boldsymbol{A}$ and noise $\boldsymbol{w}$. But it also includes non-linear inverse problems like those arising in phase-retrieval and dequantization, as well as image-to-image translation problems. In all cases, it is impossible to perfectly infer $\boldsymbol{x}$ from $\boldsymbol{y}$.

Image recovery is often posed as finding the single "best" recovery $\widehat{\boldsymbol{x}}$, which is known as a *point estimate* of $\boldsymbol{x}$ [1]. But point estimation is problematic due to the *perception-distortion tradeoff* [2], which establishes a fundamental tradeoff between distortion (defined as some distance between $\widehat{\boldsymbol{x}}$ and $\boldsymbol{x}$) and perceptual quality (defined as some distance between $\widehat{\boldsymbol{x}}$ and the set of clean images). For example, the minimum mean-squared error (MMSE) recovery $\widehat{\boldsymbol{x}}_{\text{mmse}}$ is optimal in terms of $\ell_2$ distortion, but can be unrealistically smooth. Although one could instead compute an approximation of the maximum a posteriori (MAP) estimate [3] or minimize some combination of perceptual and distortion losses, it's unclear which combination would be most appropriate.

Another major limitation with point estimation is that it's unclear how certain one can be about the recovered $\widehat{\boldsymbol{x}}$. For example, with deep-learning-based recovery, it's possible to hallucinate a nice-looking $\widehat{\boldsymbol{x}}$, but is it correct? Quantifying the uncertainty in $\widehat{\boldsymbol{x}}$ is especially important in medical applications such as MRI, where a diagnosis must be made based on the measurements $\boldsymbol{y}$. Rather than simply reporting our best guess of whether a pathology is present or absent based on $\widehat{\boldsymbol{x}}$, we might want to report the probability that the pathology is present (given all available data).

37th Conference on Neural Information Processing Systems (NeurIPS 2023).

Yet another problem with point estimation is that the estimated $\widehat{\boldsymbol{x}}$ could pose issues with fairness [4]. For example, say we are inpainting a face within an image. With a racially heterogeneous prior distribution, $\widehat{\boldsymbol{x}}_{\mathsf{mmse}}$ (being the posterior mean) will be biased towards the most predominant race. The same will be true of many other point estimates $\widehat{\boldsymbol{x}}$.

To address the aforementioned limitations of point estimation, we focus on *generating samples from the posterior distribution* $p_{\mathsf{x|y}}(\cdot|\boldsymbol{y})$, which represents the complete state-of-knowledge about $\boldsymbol{x}$ given the measurements $\boldsymbol{y}$. The posterior correctly fuses prior knowledge with measurement knowledge, thereby alleviating any concerns about fairness (assuming the data used to represent the prior is fairly chosen [4]). Furthermore, the posterior directly facilitates uncertainty quantification via, e.g., pixel-wise standard deviations or pathology detection probabilities (see Appendix A). Also, if it was important to report a single "good" recovery, then posterior sampling leads to an easy navigation of the perception-distortion tradeoff. For example, averaging $P \geq 1$ posterior samples gives a close approximation to the less-distorted-but-oversmooth $\widehat{\boldsymbol{x}}_{\mathsf{mmse}}$ with large $P$ and sharp-but-more-distorted $\widehat{\boldsymbol{x}}$ with small $P$. Additionally, posterior sampling unlocks other important capabilities such as adaptive acquisition [5] and counterfactual diagnosis [6].

Concretely, given a training dataset of image/measurement pairs $\{(\boldsymbol{x}_t, \boldsymbol{y}_t)\}_{t=1}^T$, our goal is to learn a generating function $G_{\boldsymbol{\theta}}$ that, for a new $\boldsymbol{y}$, maps random code vectors $\boldsymbol{z} \sim \mathcal{N}(\boldsymbol{0}, \boldsymbol{I})$ to posterior samples $\widehat{\boldsymbol{x}} = G_{\boldsymbol{\theta}}(\boldsymbol{z}, \boldsymbol{y}) \sim p_{\mathsf{x|y}}(\cdot|\boldsymbol{y})$. There exist several well-known approaches to this task, with recent literature focusing on conditional generative adversarial networks (cGANs) [7, 8, 9, 10], conditional variational autoencoders (cVAEs) [11, 12, 13], conditional normalizing flows (cNFs) [14, 15, 16], and score/diffusion/Langevin-based generative models [17, 18, 19, 20, 21]. Despite it being a long-standing problem, posterior image sampling remains challenging. Although score/diffusion/Langevin approaches have dominated the recent literature due to advances in accuracy and diversity, their sample-generation speeds remain orders-of-magnitude behind those of cGANs, cVAEs, and cNFs.

We choose to focus on cGANs, which are typically regarded as generating samples of high quality but low diversity. Our proposed cGAN tackles the lack-of-diversity issue using a novel regularization that consists of supervised-$\ell_1$ loss plus an adaptively weighted standard-deviation (SD) reward. This is not a heuristic choice; we prove that our regularization enforces consistency with the true posterior mean and covariance under certain conditions.

Experimentally, we demonstrate our regularized cGAN on accelerated MRI and large-scale face completion/inpainting. We consider these applications for three main reasons. First, uncertainty quantification in MRI, and fairness in face-generation, are both of paramount importance. Second, posterior-sampling has been well studied for both applications, and fine-tuned cGANs [9] and score/Langevin-based approaches [19, 20] are readily available. Third, the linear operator "$\boldsymbol{A}$" manifests very differently in these two applications,[1] which illustrates the versatility of our approach. To quantify performance, we focus on conditional Fréchet inception distance (CFID) [22], which is a principled way to quantify the difference between two high-dimensional posterior distributions, although we also report other metrics. Our results show the proposed regularized cGAN (rcGAN) outperforming existing cGANs [8, 23, 9] and the score/diffusion/Langevin approaches from [19] and [20] in all tested metrics, while generating samples $\sim 10^4$ times faster than [19, 20].

## 2 Problem formulation and background

We build on the Wasserstein cGAN framework from [8]. The goal is to design a generator network $G_{\boldsymbol{\theta}} : \mathcal{Z} \times \mathcal{Y} \to \mathcal{X}$ such that, for fixed $\boldsymbol{y}$, the random variable $\widehat{\boldsymbol{x}} = G_{\boldsymbol{\theta}}(\boldsymbol{z}, \boldsymbol{y})$ induced by $\boldsymbol{z} \sim p_{\mathsf{z}}$ has a distribution that best matches the posterior $p_{\mathsf{x|y}}(\cdot|\boldsymbol{y})$ in Wasserstein-1 distance. Here, $\mathcal{X}$, $\mathcal{Y}$, and $\mathcal{Z}$ denote the sets of $\boldsymbol{x}$, $\boldsymbol{y}$, and $\boldsymbol{z}$, respectively, and $\boldsymbol{z}$ is drawn independently of $\boldsymbol{y}$.

The Wasserstein-1 distance can be expressed as

$$W_1(p_{\mathsf{x|y}}(\cdot, \boldsymbol{y}), p_{\widehat{\mathsf{x}}|\mathsf{y}}(\cdot, \boldsymbol{y})) = \sup_{D \in L_1} \mathrm{E}_{\mathsf{x|y}}\{D(\boldsymbol{x}, \boldsymbol{y})\} - \mathrm{E}_{\widehat{\mathsf{x}}|\mathsf{y}}\{D(\widehat{\boldsymbol{x}}, \boldsymbol{y})\}, \tag{1}$$

where $L_1$ denotes functions that are 1-Lipschitz with respect to their first argument and $D : \mathcal{X} \times \mathcal{Y} \to \mathbb{R}$ is a "critic" or "discriminator" that tries to distinguish between true $\boldsymbol{x}$ and generated $\widehat{\boldsymbol{x}}$ given $\boldsymbol{y}$.

---

[1]In MRI, the forward operator acts locally in the frequency domain but globally in the pixel domain, while in inpainting, the operator acts locally in the pixel domain but globally in the frequency domain.

Since we want the method to work for typical values of $\boldsymbol{y}$, we define a loss by taking an expectation of (1) over $\boldsymbol{y} \sim p_{\mathsf{y}}$. Since the expectation commutes with the supremum in (1), we have [8]

$$\mathrm{E}_{\mathsf{y}}\{W_1(p_{\mathsf{x}|\mathsf{y}}(\cdot, \boldsymbol{y}), p_{\widehat{\mathsf{x}}|\mathsf{y}}(\cdot, \boldsymbol{y}))\} = \sup_{D \in L_1} \mathrm{E}_{\mathsf{x},\mathsf{y}}\{D(\boldsymbol{x}, \boldsymbol{y})\} - \mathrm{E}_{\widehat{\mathsf{x}},\mathsf{y}}\{D(\widehat{\boldsymbol{x}}, \boldsymbol{y})\} \tag{2}$$

$$= \sup_{D \in L_1} \mathrm{E}_{\mathsf{x},\mathsf{z},\mathsf{y}}\{D(\boldsymbol{x}, \boldsymbol{y}) - D(G_{\boldsymbol{\theta}}(\boldsymbol{z}, \boldsymbol{y}), \boldsymbol{y})\}. \tag{3}$$

In practice, $D$ is implemented by a neural network $D_{\boldsymbol{\phi}}$ with parameters $\boldsymbol{\phi}$, and $(\boldsymbol{\theta}, \boldsymbol{\phi})$ are trained by alternately minimizing

$$\mathcal{L}_{\mathsf{adv}}(\boldsymbol{\theta}, \boldsymbol{\phi}) \triangleq \mathrm{E}_{\mathsf{x},\mathsf{z},\mathsf{y}}\{D_{\boldsymbol{\phi}}(\boldsymbol{x}, \boldsymbol{y}) - D_{\boldsymbol{\phi}}(G_{\boldsymbol{\theta}}(\boldsymbol{z}, \boldsymbol{y}), \boldsymbol{y})\} \tag{4}$$

with respect to $\boldsymbol{\theta}$ and minimizing $-\mathcal{L}_{\mathsf{adv}}(\boldsymbol{\theta}, \boldsymbol{\phi}) + \mathcal{L}_{\mathsf{gp}}(\boldsymbol{\phi})$ with respect to $\boldsymbol{\phi}$, where $\mathcal{L}_{\mathsf{gp}}(\boldsymbol{\phi})$ is a gradient penalty that is used to encourage $D_{\boldsymbol{\phi}} \in L_1$ [24]. Furthemore, the expectation over $\boldsymbol{x}$ and $\boldsymbol{y}$ in (4) is replaced in practice by a sample average over the training examples $\{(\boldsymbol{x}_t, \boldsymbol{y}_t)\}_{t=1}^T$.

One of the main challenges with the cGAN framework in image recovery problems is that, for each measurement example $\boldsymbol{y}_t$, there is only a single image example $\boldsymbol{x}_t$. Thus, with the previously described training methodology, there is no incentive for the generator to produce diverse samples $G(\boldsymbol{z}, \boldsymbol{y})|_{\boldsymbol{z} \sim p_{\mathsf{z}}}$ for a fixed $\boldsymbol{y}$. This can lead to the generator learning to ignore the code vector $\boldsymbol{z}$, which causes a form of "mode collapse."

Although issues with stability and mode collapse are also present in *unconditional* GANs (uGANs) or discretely conditioned cGANs [25], the causes are fundamentally different than in continuously conditioned cGANs like ours. With continuously conditioned cGANs, there is only *one* example of a valid $\boldsymbol{x}_t$ for each given $\boldsymbol{y}_t$, whereas with uGANs there are many $\boldsymbol{x}_t$ and with discretely conditioned cGANs there are many $\boldsymbol{x}_t$ for each conditioning class. As a result, most strategies that are used to combat mode-collapse in uGANs [26, 27, 28] are not well suited to cGANs. For example, mini-batch discrimination strategies like MBSD [29], where the discriminator aims to distinguish a mini-batch of true samples $\{\boldsymbol{x}_t\}$ from a mini-batch of generated samples $\{\widehat{\boldsymbol{x}}_t\}$, don't work with cGANs because the posterior statistics are very different than the prior statistics.

To combat mode collapse in cGANs, Adler & Öktem [8] proposed to use a three-input discriminator $D_{\boldsymbol{\phi}}^{\mathsf{adler}} : \mathcal{X} \times \mathcal{X} \times \mathcal{Y} \to \mathbb{R}$ and replace $\mathcal{L}_{\mathsf{adv}}$ from (4) with the loss

$$\mathcal{L}_{\mathsf{adv}}^{\mathsf{adler}}(\boldsymbol{\theta}, \boldsymbol{\phi}) \triangleq \mathrm{E}_{\mathsf{x},\mathsf{z}_1,\mathsf{z}_2,\mathsf{y}} \left\{ \tfrac{1}{2} D_{\boldsymbol{\phi}}^{\mathsf{adler}}(\boldsymbol{x}, G_{\boldsymbol{\theta}}(\boldsymbol{z}_1, \boldsymbol{y}), \boldsymbol{y}) + \tfrac{1}{2} D_{\boldsymbol{\phi}}^{\mathsf{adler}}(G_{\boldsymbol{\theta}}(\boldsymbol{z}_2, \boldsymbol{y}), \boldsymbol{x}, \boldsymbol{y}) \right.$$
$$\left. - D_{\boldsymbol{\phi}}^{\mathsf{adler}}(G_{\boldsymbol{\theta}}(\boldsymbol{z}_1, \boldsymbol{y}), G_{\boldsymbol{\theta}}(\boldsymbol{z}_2, \boldsymbol{y}), \boldsymbol{y}) \right\}, \tag{5}$$

which rewards variation between the first and second inputs to $D_{\boldsymbol{\phi}}^{\mathsf{adler}}$. They then proved that minimizing $\mathcal{L}_{\mathsf{adv}}^{\mathsf{adler}}$ in place of $\mathcal{L}_{\mathsf{adv}}$ does not compromise the Wasserstein cGAN objective, i.e., $\arg\min_{\boldsymbol{\theta}} \mathcal{L}_{\mathsf{adv}}^{\mathsf{adler}}(\boldsymbol{\theta}, \boldsymbol{\phi}) = \arg\min_{\boldsymbol{\theta}} \mathcal{L}_{\mathsf{adv}}(\boldsymbol{\theta}, \boldsymbol{\phi})$. As we show in Section 4, this approach does prevent complete mode collapse, but it leaves much room for improvement.

## 3 Proposed method

### 3.1 Proposed regularization: supervised-$\ell_1$ plus SD reward

We now propose a novel cGAN regularization framework. To train the generator, we propose to solve

$$\arg\min_{\boldsymbol{\theta}} \{\beta_{\mathsf{adv}} \mathcal{L}_{\mathsf{adv}}(\boldsymbol{\theta}, \boldsymbol{\phi}) + \mathcal{L}_{1,\mathsf{SD},P_{\mathsf{train}}}(\boldsymbol{\theta}, \beta_{\mathsf{SD}})\} \tag{6}$$

with appropriately chosen $\beta_{\mathsf{adv}}, \beta_{\mathsf{SD}} > 0$ and $P_{\mathsf{train}} \geq 2$, where the regularizer

$$\mathcal{L}_{1,\mathsf{SD},P_{\mathsf{train}}}(\boldsymbol{\theta}, \beta_{\mathsf{SD}}) \triangleq \mathcal{L}_{1,P_{\mathsf{train}}}(\boldsymbol{\theta}) - \beta_{\mathsf{SD}} \mathcal{L}_{\mathsf{SD},P_{\mathsf{train}}}(\boldsymbol{\theta}) \tag{7}$$

is constructed from the $P_{\mathsf{train}}$-sample supervised-$\ell_1$ loss and standard-deviation (SD) reward terms

$$\mathcal{L}_{1,P_{\mathsf{train}}}(\boldsymbol{\theta}) \triangleq \mathrm{E}_{\mathsf{x},\mathsf{z}_1,\ldots,\mathsf{z}_P,\mathsf{y}} \left\{ \|\boldsymbol{x} - \widehat{\boldsymbol{x}}_{(P_{\mathsf{train}})}\|_1 \right\} \tag{8}$$

$$\mathcal{L}_{\mathsf{SD},P_{\mathsf{train}}}(\boldsymbol{\theta}) \triangleq \sqrt{\tfrac{\pi}{2 P_{\mathsf{train}}(P_{\mathsf{train}}-1)}} \sum_{i=1}^{P_{\mathsf{train}}} \mathrm{E}_{\mathsf{z}_1,\ldots,\mathsf{z}_P,\mathsf{y}} \left\{ \|\widehat{\boldsymbol{x}}_i - \widehat{\boldsymbol{x}}_{(P_{\mathsf{train}})}\|_1 \right\}, \tag{9}$$

and where $\{\widehat{\boldsymbol{x}}_i\}$ denote the generated samples and $\widehat{\boldsymbol{x}}_{(P)}$ their $P$-sample average:

$$\widehat{\boldsymbol{x}}_i \triangleq G_{\boldsymbol{\theta}}(\boldsymbol{z}_i, \boldsymbol{y}), \qquad \widehat{\boldsymbol{x}}_{(P)} \triangleq \tfrac{1}{P} \sum_{i=1}^P \widehat{\boldsymbol{x}}_i. \tag{10}$$

The use of supervised-$\ell_1$ loss and SD reward in (7) is not heuristic. As shown in Proposition 3.1, it encourages the samples $\{\widehat{\boldsymbol{x}}_i\}$ to match the true posterior in both mean and covariance.

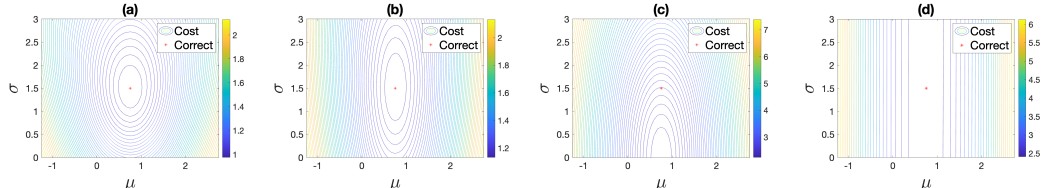

Figure 1: The contours show the regularizer value versus $\boldsymbol{\theta} = [\mu, \sigma]^\top$ for four different regularizers: (a) supervised-$\ell_1$ plus SD reward with $\beta_{\mathsf{SD}} = \beta_{\mathsf{SD}}^{\mathcal{N}}$ at $P_{\mathsf{train}} = 2$, (b) supervised-$\ell_1$ plus SD reward with $\beta_{\mathsf{SD}} = \beta_{\mathsf{SD}}^{\mathcal{N}}$ at $P_{\mathsf{train}} = 8$, (c) supervised-$\ell_2$ at $P_{\mathsf{train}} = 8$, and (d) supervised-$\ell_2$ plus variance reward at $P_{\mathsf{train}} = 8$. The red star shows the true posterior parameters $[\mu_0, \sigma_0]^\top$.

**Proposition 3.1.** *Suppose $P_{\mathsf{train}} \geq 2$ and $\boldsymbol{\theta}$ has complete control over the $\boldsymbol{y}$-conditional mean and covariance of $\widehat{\boldsymbol{x}}_i$. Then the parameters $\boldsymbol{\theta}_* = \arg\min_{\boldsymbol{\theta}} \mathcal{L}_{1,\mathsf{SD},P_{\mathsf{train}}}(\boldsymbol{\theta}, \beta_{\mathsf{SD}}^{\mathcal{N}})$ with*

$$\beta_{\mathsf{SD}}^{\mathcal{N}} \triangleq \sqrt{\frac{2}{\pi P_{\mathsf{train}}(P_{\mathsf{train}}+1)}} \tag{11}$$

*yield generated statistics*

$$\mathrm{E}_{\mathsf{z}_i|\mathsf{y}}\{\widehat{\boldsymbol{x}}_i(\boldsymbol{\theta}_*)|\boldsymbol{y}\} = \mathrm{E}_{\mathsf{x}|\mathsf{y}}\{\boldsymbol{x}|\boldsymbol{y}\} = \widehat{\boldsymbol{x}}_{\mathsf{mmse}} \tag{12a}$$

$$\mathrm{Cov}_{\mathsf{z}_i|\mathsf{y}}\{\widehat{\boldsymbol{x}}_i(\boldsymbol{\theta}_*)|\boldsymbol{y}\} = \mathrm{Cov}_{\mathsf{x}|\mathsf{y}}\{\boldsymbol{x}|\boldsymbol{y}\} \tag{12b}$$

*when the elements of $\widehat{\boldsymbol{x}}_i$ and $\boldsymbol{x}$ are independent Gaussian conditioned on $\boldsymbol{y}$. Thus, minimizing $\mathcal{L}_{1,\mathsf{SD},P_{\mathsf{train}}}$ encourages the $\boldsymbol{y}$-conditional mean and covariance of $\widehat{\boldsymbol{x}}_i$ to match those of the true $\boldsymbol{x}$.*

See Appendix B for a proof. In imaging applications, $\widehat{\boldsymbol{x}}_i$ and $\boldsymbol{x}$ may not be independent Gaussian conditioned on $\boldsymbol{y}$, and so the value of $\beta_{\mathsf{SD}}$ in (11) may not be appropriate. Thus we propose a method to automatically tune $\beta_{\mathsf{SD}}$ in Section 3.2.

Figure 1 shows a toy example with parameters $\boldsymbol{\theta} = [\mu, \sigma]^\top$, generator $G_{\boldsymbol{\theta}}(z, y) = \mu + \sigma z$, and $z \sim \mathcal{N}(0, 1)$, giving generated posterior $p_{\widehat{\mathsf{x}}|\mathsf{y}}(x|y) = \mathcal{N}(x; \mu, \sigma^2)$. Assuming the true $p_{\mathsf{x}|\mathsf{y}}(x|y) = \mathcal{N}(x; \mu_0, \sigma_0^2)$, Figs. 1(a)-(b) show that, by minimizing the proposed $\mathcal{L}_{1,\mathsf{SD},P_{\mathsf{train}}}(\boldsymbol{\theta}, \beta_{\mathsf{SD}}^{\mathcal{N}})$ regularization over $\boldsymbol{\theta} = [\mu, \sigma]^\top$ for any $P_{\mathsf{train}} \geq 2$, we recover the true $\boldsymbol{\theta}_0 = [\mu_0, \sigma_0]^\top$. They also show that the cost function steepens as $P_{\mathsf{train}}$ decreases, with agrees with our empirical finding that $P_{\mathsf{train}} = 2$ tends to work best in practice.

We note that regularizing a cGAN with supervised-$\ell_1$ loss alone is not new; see, e.g., [7]. In fact, the use of supervised-$\ell_1$ loss is often preferred over $\ell_2$ in image recovery because it results in sharper, more visually pleasing results [30]. But regularizing a cGAN using supervised-$\ell_1$ loss *alone* can push the generator towards mode collapse, for reasons described below. For example, in [7], $\ell_1$-induced mode collapse led the authors to use dropout to induce generator variation, instead of random $\boldsymbol{z}_i$.

**Why not supervised-$\ell_2$ regularization?** One may wonder: Why regularize using supervised-$\ell_1$ loss plus an SD reward in (7) and not a more conventional choice like supervised-$\ell_2$ loss plus a variance reward, or even supervised-$\ell_2$ loss alone? We start by discussing the latter.

The use of supervised-$\ell_2$ regularization in a cGAN can be found in [7]. In this case, to train the generator, one would aim to solve $\arg\min_{\boldsymbol{\theta}}\{\mathcal{L}_{\mathsf{adv}}(\boldsymbol{\theta}, \phi) + \lambda \mathcal{L}_2(\boldsymbol{\theta})\}$ with some $\lambda > 0$ and

$$\mathcal{L}_2(\boldsymbol{\theta}) \triangleq \mathrm{E}_{\mathsf{x},\mathsf{y}}\left\{\|\boldsymbol{x} - \mathrm{E}_{\mathsf{z}}\{G_{\boldsymbol{\theta}}(\boldsymbol{z}, \boldsymbol{y})\}\|_2^2\right\}. \tag{13}$$

Ohayon et al. [23] revived this idea for the explicit purpose of fighting mode collapse. In practice, however, the $\mathrm{E}_{\mathsf{z}}$ term in (13) must be implemented by a finite-sample average, giving

$$\mathcal{L}_{2,P_{\mathsf{train}}}(\boldsymbol{\theta}) \triangleq \mathrm{E}_{\mathsf{x},\mathsf{z}_1,\ldots,\mathsf{z}_P,\mathsf{y}}\left\{\left\|\boldsymbol{x} - \frac{1}{P_{\mathsf{train}}}\sum_{i=1}^{P_{\mathsf{train}}} G_{\boldsymbol{\theta}}(\boldsymbol{z}_i, \boldsymbol{y})\right\|_2^2\right\} \tag{14}$$

for some $P_{\mathsf{train}} \geq 2$. For example, Ohayon's implementation [31] used $P_{\mathsf{train}} = 8$. As we show in Proposition 3.2, $\mathcal{L}_{2,P_{\mathsf{train}}}$ *induces* mode collapse rather than prevents it.

**Proposition 3.2.** *Say $P_{\mathsf{train}}$ is finite and $\boldsymbol{\theta}$ has complete control over the $\boldsymbol{y}$-conditional mean and covariance of $\widehat{\boldsymbol{x}}_i$. Then the parameters $\boldsymbol{\theta}_* = \arg\min_{\boldsymbol{\theta}} \mathcal{L}_{2,P_{\mathsf{train}}}(\boldsymbol{\theta})$ yield generated statistics*

$$\mathrm{E}_{\mathsf{z}_i|\mathsf{y}}\{\widehat{\boldsymbol{x}}_i(\boldsymbol{\theta}_*)|\boldsymbol{y}\} = \mathrm{E}_{\mathsf{x}|\mathsf{y}}\{\boldsymbol{x}|\boldsymbol{y}\} = \widehat{\boldsymbol{x}}_{\mathsf{mmse}} \tag{15a}$$

$$\mathrm{Cov}_{\mathsf{z}_i|\mathsf{y}}\{\widehat{\boldsymbol{x}}_i(\boldsymbol{\theta}_*)|\boldsymbol{y}\} = \boldsymbol{0}. \tag{15b}$$

*Thus, minimizing $\mathcal{L}_{2,P_{\mathsf{train}}}$ encourages mode collapse.*

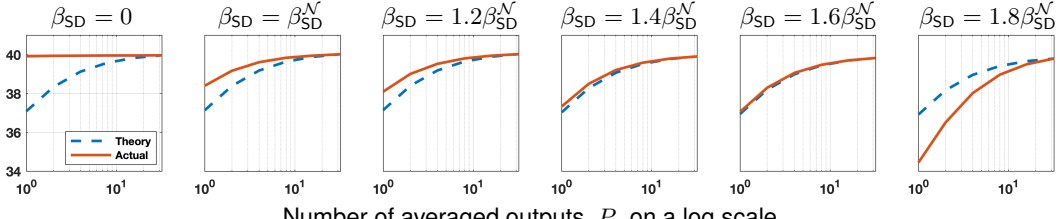

Figure 2: Example PSNR of $\widehat{x}_{(P)}$ versus $P$, the number of averaged outputs, for several training $\beta_{\mathsf{SD}}$ and MRI recovery at $R = 4$. Also shown is the theoretical behavior for true-posterior samples.

The proof (see Appendix C) follows from the bias-variance decomposition of (14), i.e.,

$$
\begin{aligned}
&\mathcal{L}_{2,P_{\text{train}}}(\boldsymbol{\theta}) \\
&= \mathrm{E}_{\mathsf{y}}\left\{\|\widehat{x}_{\mathsf{mmse}} - \mathrm{E}_{\mathsf{z}_i|\mathsf{y}}\{\widehat{x}_i(\boldsymbol{\theta})|\boldsymbol{y}\}\|_2^2 + \tfrac{1}{P_{\text{train}}}\operatorname{tr}[\operatorname{Cov}_{\mathsf{z}_i|\mathsf{y}}\{\widehat{x}_i(\boldsymbol{\theta})|\boldsymbol{y}\}] + \mathrm{E}_{\mathsf{x}|\mathsf{y}}\{\|\boldsymbol{e}_{\mathsf{mmse}}\|_2^2|\boldsymbol{y}\}\right\},
\end{aligned}
\quad (16)
$$

where $\boldsymbol{e}_{\mathsf{mmse}} \triangleq \boldsymbol{x} - \boldsymbol{x}_{\mathsf{mmse}}$ is the MMSE error. Figure 1(c) shows that $\mathcal{L}_{2,P_{\text{train}}}$ regularization causes mode collapse in the toy example, and Section 4.2 shows that it causes mode collapse in MRI.

**Why not supervised $\ell_2$ plus a variance reward?** To mitigate $\mathcal{L}_{2,P_{\text{train}}}$'s incentive for mode-collapse, the second term in (16) could be canceled using a variance reward, giving

$$
\mathcal{L}_{2,\mathsf{var},P_{\text{train}}}(\boldsymbol{\theta}) \triangleq \mathcal{L}_{2,P_{\text{train}}}(\boldsymbol{\theta}) - \tfrac{1}{P_{\text{train}}}\mathcal{L}_{\mathsf{var},P_{\text{train}}}(\boldsymbol{\theta}) \quad (17)
$$

$$
\text{with } \mathcal{L}_{\mathsf{var},P_{\text{train}}}(\boldsymbol{\theta}) \triangleq \tfrac{1}{P_{\text{train}}-1}\sum_{i=1}^{P_{\text{train}}}\mathrm{E}_{\mathsf{z}_1,\ldots,\mathsf{z}_P,\mathsf{y}}\{\|\widehat{x}_i(\boldsymbol{\theta}) - \widehat{x}_{(P)}(\boldsymbol{\theta})\|_2^2\}. \quad (18)
$$

since Appendix D shows that $\mathcal{L}_{\mathsf{var},P_{\text{train}}}(\boldsymbol{\theta})$ is an unbiased estimator of the posterior trace-covariance:

$$
\mathcal{L}_{\mathsf{var},P_{\text{train}}}(\boldsymbol{\theta}) = \mathrm{E}_{\mathsf{y}}\{\operatorname{tr}[\operatorname{Cov}_{\mathsf{z}_i|\mathsf{y}}\{\widehat{x}_i(\boldsymbol{\theta})|\boldsymbol{y}\}]\} \text{ for any } P_{\text{train}} \geq 2. \quad (19)
$$

However, the resulting $\mathcal{L}_{2,\mathsf{var},P_{\text{train}}}$ regularizer in (17) does not encourage the generated covariance to match the *true* posterior covariance, unlike the proposed $\mathcal{L}_{1,\mathsf{SD},P_{\text{train}}}$ regularizer in (7) (recall Proposition 3.1). For the toy example, this behavior is visible in Fig. 1(d).

### 3.2 Auto-tuning of SD reward weight $\beta_{\mathsf{SD}}$

We now propose a method to auto-tune $\beta_{\mathsf{SD}}$ in (7) for a given training dataset. Our approach is based on the observation that larger values of $\beta_{\mathsf{SD}}$ tend to yield samples $\widehat{x}_i$ with more variation. But more variation is not necessarily better; we want samples with the correct amount of variation. To assess the correct amount of variation, we compare the expected $\ell_2$ error of the $P$-sample average $\widehat{x}_{(P)}$ to that of $\widehat{x}_{(1)}$. When $\{\widehat{x}_i\}$ are true-posterior samples, these errors follow a particular relationship, as established by Proposition 3.3 below (see Appendix E for a proof).

**Proposition 3.3.** *Say $\widehat{x}_i \sim p_{\mathsf{x}|\mathsf{y}}(\cdot|\boldsymbol{y})$ are independent samples of the true posterior and, for any $P \geq 1$, their $P$-sample average is $\widehat{x}_{(P)} \triangleq \tfrac{1}{P}\sum_{i=1}^{P}\widehat{x}_i$. Then*

$$
\mathcal{E}_P \triangleq \mathrm{E}\{\|\widehat{x}_{(P)} - \boldsymbol{x}\|_2^2|\boldsymbol{y}\} = \tfrac{P+1}{P}\mathcal{E}_{\mathsf{mmse}}, \quad \text{which implies} \quad \tfrac{\mathcal{E}_1}{\mathcal{E}_P} = \tfrac{2P}{P+1}. \quad (20)
$$

Experimentally we find that $\mathcal{E}_1/\mathcal{E}_P$ grows with the SD reward weight $\beta_{\mathsf{SD}}$. (See Fig. 2.) Thus, we propose to adjust $\beta_{\mathsf{SD}}$ so that the observed SNR-gain ratio $\mathcal{E}_1/\mathcal{E}_{P_{\mathsf{val}}}$ matches the correct ratio $(2P_{\mathsf{val}})/(P_{\mathsf{val}} + 1)$ from (20), for some $P_{\mathsf{val}} \geq 2$ that does not need to match $P_{\text{train}}$. (We use $P_{\mathsf{val}} = 8$ in Section 4.) In particular, at each training epoch $\tau$, we approximate $\mathcal{E}_{P_{\mathsf{val}}}$ and $\mathcal{E}_1$ as follows:

$$
\widehat{\mathcal{E}}_{P_{\mathsf{val}},\tau} \triangleq \tfrac{1}{V}\sum_{v=1}^{V}\|\tfrac{1}{P_{\mathsf{val}}}\sum_{i=1}^{P_{\mathsf{val}}}G_{\boldsymbol{\theta}_\tau}(\boldsymbol{z}_{i,v}, \boldsymbol{y}_v) - \boldsymbol{x}_v\|_2^2 \quad (21)
$$

$$
\widehat{\mathcal{E}}_{1,\tau} \triangleq \tfrac{1}{V}\sum_{v=1}^{V}\|G_{\boldsymbol{\theta}_\tau}(\boldsymbol{z}_{1,v}, \boldsymbol{y}_v) - \boldsymbol{x}_v\|_2^2, \quad (22)
$$

with validation set $\{(\boldsymbol{x}_v, \boldsymbol{y}_v)\}_{v=1}^{V}$ and i.i.d. codes $\{\boldsymbol{z}_{i,v}\}_{i=1}^{P_{\mathsf{val}}}$. We update $\beta_{\mathsf{SD}}$ using gradient descent:

$$
\beta_{\mathsf{SD},\tau+1} = \beta_{\mathsf{SD},\tau} - \mu_{\mathsf{SD}} \cdot ([\widehat{\mathcal{E}}_{1,\tau}/\widehat{\mathcal{E}}_{P_{\mathsf{val}},\tau}]_{\mathsf{dB}} - [2P_{\mathsf{val}}/(P_{\mathsf{val}}+1)]_{\mathsf{dB}})\beta_{\mathsf{SD}}^{\mathcal{N}} \text{ for } \tau = 0, 1, 2, \ldots \quad (23)
$$

with $\beta_{\mathsf{SD},0} = \beta_{\mathsf{SD}}^{\mathcal{N}}$, some $\mu_{\mathsf{SD}} > 0$, and $[x]_{\mathsf{dB}} \triangleq 10\log_{10}(x)$.

# 4 Numerical experiments

## 4.1 Conditional Fréchet inception distance

As previously stated, our goal is to train a generator $G_{\boldsymbol{\theta}}$ so that, for typical fixed values of $\boldsymbol{y}$, the generated distribution $p_{\widehat{\mathsf{x}}|\mathsf{y}}(\cdot|\boldsymbol{y})$ matches the true posterior $p_{\mathsf{x}|\mathsf{y}}(\cdot|\boldsymbol{y})$. It is essential to have a quantitative metric for evaluating performance with respect to this goal. For example, it is not enough that the generated samples are "accurate" in the sense of being close to the ground truth, nor is it enough that they are "diverse" according to some heuristically chosen metric.

We quantify posterior-approximation quality using the conditional Fréchet inception distance (CFID) [22], a computationally efficient approximation to the conditional Wasserstein distance

$$\mathrm{CWD} \triangleq \mathrm{E}_{\mathsf{y}}\{W_2(p_{\mathsf{x}|\mathsf{y}}(\cdot, \boldsymbol{y}), p_{\widehat{\mathsf{x}}|\mathsf{y}}(\cdot, \boldsymbol{y}))\}. \tag{24}$$

In (24), $W_2(p_{\mathsf{a}}, p_{\mathsf{b}})$ denotes the Wasserstein-2 distance between distributions $p_{\mathsf{a}}$ and $p_{\mathsf{b}}$, defined as

$$W_2(p_{\mathsf{a}}, p_{\mathsf{b}}) \triangleq \min_{p_{\mathsf{a},\mathsf{b}} \in \Pi(p_{\mathsf{a}}, p_{\mathsf{b}})} \mathrm{E}_{\mathsf{a},\mathsf{b}}\{\|\boldsymbol{a} - \boldsymbol{b}\|_2^2\}, \tag{25}$$

where $\Pi(p_{\mathsf{a}}, p_{\mathsf{b}}) \triangleq \{p_{\mathsf{a},\mathsf{b}} : p_{\mathsf{a}} = \int p_{\mathsf{a},\mathsf{b}} \, \mathrm{d}\boldsymbol{b} \text{ and } p_{\mathsf{b}} = \int p_{\mathsf{a},\mathsf{b}} \, \mathrm{d}\boldsymbol{a}\}$ denotes the set of joint distributions $p_{\mathsf{a},\mathsf{b}}$ with prescribed marginals $p_{\mathsf{a}}$ and $p_{\mathsf{b}}$. Similar to how FID [32]—a popular uGAN metric—is computed, CFID approximates CWD (24) as follows: i) the random vectors $\boldsymbol{x}$, $\widehat{\boldsymbol{x}}$, and $\boldsymbol{y}$ are replaced by low-dimensional embeddings $\underline{\boldsymbol{x}}$, $\underline{\widehat{\boldsymbol{x}}}$, and $\underline{\boldsymbol{y}}$, generated by the convolutional layers of a deep network, and ii) the embedding distributions $p_{\underline{\mathsf{x}}|\underline{\mathsf{y}}}$ and $p_{\underline{\widehat{\mathsf{x}}}|\underline{\mathsf{y}}}$ are approximated by multivariate Gaussians. More details on CFID are given in Appendix F.

## 4.2 MRI experiments

We consider parallel MRI recovery, where the goal is to recover a complex-valued multicoil image $\boldsymbol{x}$ from zero-filled measurements $\boldsymbol{y}$ (see Appendix G for details).

**Data.** For training data $\{\boldsymbol{x}_t\}$, we used the first 8 slices of all fastMRI [33] T2 brain training volumes with at least 8 coils, cropping to $384 \times 384$ pixels and compressing to 8 virtual coils [34], yielding 12 200 training images. Using the same procedure, 2 376 testing and 784 validation images were obtained from the fastMRI T2 brain testing volumes. From the 2 376 testing images, 72 were randomly selected to evaluate the Langevin technique [19] in order to limit its sample generation to 6 days. To create the measurement $\boldsymbol{y}_t$, we transformed $\boldsymbol{x}_t$ to the Fourier domain, sampled using pseudo-random GRO patterns [35] at acceleration $R = 4$ and $R = 8$, and Fourier-transformed the zero-filled k-space measurements back to the (complex, multicoil) image domain.

**Architecture.** We use a UNet [36] for our generator and a standard CNN for our generator, along with data-consistency as in Appendix H. Architecture and training details are given in Appendix I.

**Competitors.** We compare our cGAN to the Adler and Öktem's cGAN [8], Ohayon et al.'s cGAN [23], Jalal et al.'s fastMRI Langevin approach [19], and Sriram et al.'s E2E-VarNet [37]. The cGAN from [8] uses generator loss $\beta_{\mathsf{adv}}\mathcal{L}_{\mathsf{adv}}^{\mathsf{adler}}(\boldsymbol{\theta}, \boldsymbol{\phi})$ and discriminator loss $-\mathcal{L}_{\mathsf{adv}}^{\mathsf{adler}}(\boldsymbol{\theta}, \boldsymbol{\phi}) + \alpha_1\mathcal{L}_{\mathsf{gp}}(\boldsymbol{\phi}) + \alpha_2\mathcal{L}_{\mathsf{drift}}(\boldsymbol{\phi})$, while the cGAN from [23] uses generator loss $\beta_{\mathsf{adv}}\mathcal{L}_{\mathsf{adv}}(\boldsymbol{\theta}, \boldsymbol{\phi}) + \mathcal{L}_{2,P}(\boldsymbol{\theta})$ and discriminator loss $-\mathcal{L}_{\mathsf{adv}}(\boldsymbol{\theta}, \boldsymbol{\phi}) + \alpha_1\mathcal{L}_{\mathsf{gp}}(\boldsymbol{\phi}) + \alpha_2\mathcal{L}_{\mathsf{drift}}(\boldsymbol{\phi})$. Each used the value of $\beta_{\mathsf{adv}}$ specified in the original paper. All cGANs used the same generator and discriminator architectures, except that [8] used extra discriminator input channels to facilitate the 3-input loss (5). For the fastMRI Langevin approach [19], we did not modify the authors' implementation in [38] except to use the GRO sampling mask. For the E2E-VarNet [37], we use the same training procedure and hyperparameters outlined in [19] except that we use the GRO sampling mask.

**Testing.** To evaluate performance, we converted the multicoil outputs $\widehat{\boldsymbol{x}}_i$ to complex-valued images using SENSE-based coil combining [39] with ESPIRiT-estimated [40] coil sensitivity maps, as described in Appendix G. Magnitude images were used to compute performance metrics.

**Results.** Table 1 shows CFID, FID, APSD $\triangleq \left(\frac{1}{NP} \sum_{i=1}^P \|\widehat{\boldsymbol{x}}_{(P)} - \widehat{\boldsymbol{x}}_i\|^2\right)^{1/2}$, and 4-sample generation time at $R \in \{4, 8\}$. (C)FID was computed using VGG-16 (not Inception-v3) to better align with radiologists' perceptions [41]. As previously described, the Langevin method was evaluated using only 72 test images. Because CFID is biased at small sample sizes [22], we re-evaluated the other

Table 1: Average MRI results at $R \in \{4, 8\}$. CFID[1], FID, and APSD used 72 test samples and $P = 32$, CFID[2] used 2 376 test samples and $P = 8$, and CFID[3] used all 14 576 samples and $P = 1$

| | R = 4 | | | | | | R = 8 | | | | | |
|---|---|---|---|---|---|---|---|---|---|---|---|---|
| Model | CFID[1]↓ | CFID[2]↓ | CFID[3]↓ | FID↓ | APSD | Time (4)↓ | CFID[1]↓ | CFID[2]↓ | CFID[3]↓ | FID↓ | APSD | Time (4)↓ |
| E2E-VarNet (Sriram et al. [37]) | 7.47 | 6.99 | 6.61 | 8.84 | 0.0 | 310ms | 7.82 | 6.81 | 6.31 | 8.40 | 0.0 | 316ms |
| Langevin (Jalal et al. [19]) | 5.29 | - | - | 6.12 | 5.9e-6 | 14 min | 7.34 | - | - | 14.32 | 7.6e-6 | 14 min |
| cGAN (Adler & Öktem [8]) | 6.39 | 4.27 | 3.82 | 5.25 | 3.9e-6 | **217 ms** | 10.10 | 6.30 | 5.72 | 10.77 | 7.7e-6 | **217 ms** |
| cGAN (Ohayon et al. [23]) | 4.06 | 3.27 | 2.95 | 6.45 | 7.2e-8 | **217 ms** | 6.04 | 4.59 | 4.27 | 11.05 | 7.7e-7 | **217 ms** |
| cGAN (Ours) | **3.10** | **1.54** | **1.29** | **3.75** | 3.8e-6 | **217 ms** | **4.87** | **2.23** | **1.79** | **7.72** | 7.6e-6 | **217 ms** |

Table 2: Average PSNR, SSIM, LPIPS, and DISTS of $\widehat{\boldsymbol{x}}_{(P)}$ versus $P$ for $R = 4$ MRI

| | PSNR↑ | | | | | | SSIM↑ | | | | | |
|---|---|---|---|---|---|---|---|---|---|---|---|---|
| Model | P=1 | P=2 | P=4 | P=8 | P=16 | P=32 | P=1 | P=2 | P=4 | P=8 | P=16 | P=32 |
| E2E-VarNet (Sriram et al. [37]) | **39.93** | - | - | - | - | - | **0.9641** | - | - | - | - | - |
| Langevin (Jalal et al. [19]) | 36.04 | 37.02 | 37.65 | 37.99 | 38.17 | 38.27 | 0.8989 | 0.9138 | 0.9218 | 0.9260 | 0.9281 | 0.9292 |
| cGAN (Adler & Öktem [8]) | 35.63 | 36.64 | 37.24 | 37.56 | 37.73 | 37.82 | 0.9330 | 0.9445 | 0.9478 | 0.9480 | 0.9477 | 0.9473 |
| cGAN (Ohayon et al. [23]) | 39.44 | 39.46 | 39.46 | 39.47 | 39.47 | 39.47 | 0.9558 | 0.9546 | 0.9539 | 0.9535 | 0.9533 | 0.9532 |
| cGAN (Ours) | 36.96 | 38.14 | 38.84 | 39.24 | 39.44 | 39.55 | 0.9440 | 0.9526 | 0.9544 | 0.9542 | 0.9537 | 0.9533 |

| | LPIPS↓ | | | | | | DISTS↓ | | | | | |
|---|---|---|---|---|---|---|---|---|---|---|---|---|
| Model | P=1 | P=2 | P=4 | P=8 | P=16 | P=32 | P=1 | P=2 | P=4 | P=8 | P=16 | P=32 |
| E2E-VarNet (Sriram et al. [37]) | 0.0316 | - | - | - | - | - | 0.0859 | - | - | - | - | - |
| Langevin (Jalal et al. [19]) | 0.0545 | 0.0394 | 0.0336 | 0.0320 | 0.0317 | 0.0316 | 0.1116 | 0.0921 | 0.0828 | 0.0793 | 0.0781 | 0.0777 |
| cGAN (Adler & Öktem [8]) | 0.0285 | 0.0255 | 0.0273 | 0.0298 | 0.0316 | 0.0327 | 0.0972 | 0.0857 | 0.0878 | 0.0930 | 0.0967 | 0.0990 |
| cGAN (Ohayon et al. [23]) | 0.0245 | 0.0247 | 0.0248 | 0.0249 | 0.0249 | 0.0249 | 0.0767 | 0.0790 | 0.0801 | 0.0807 | 0.0810 | 0.0811 |
| cGAN (Ours) | 0.0175 | **0.0164** | 0.0188 | 0.0216 | 0.0235 | 0.0245 | **0.0546** | 0.0563 | 0.0667 | 0.0755 | 0.0809 | 0.0837 |

methods using all 2 376 test images, and again using all 14 576 training and test images. Table 1 shows that our approach gave significantly better CFID and FID than the competitors. Also, the APSD of Ohayon et al.'s cGAN was an order-of-magnitude smaller than the others, indicating mode collapse. The cGANs generated samples 3 800 times faster than the Langevin approach from [19].

Tables 2 and 3 show PSNR, SSIM, LPIPS [42], and DISTS [43] for the $P$-sample average $\widehat{\boldsymbol{x}}_{(P)}$ at $P \in \{1, 2, 4, 8, 16, 32\}$ and $R \in \{4, 8\}$, respectively. While the E2E-VarNet achieves the best PSNR at $R \in \{4, 8\}$ and the best SSIM at $R = 4$, the proposed cGAN achieves the best LPIPS and DISTS performances at $R \in \{4, 8\}$ when $P = 2$ and the best SSIM at $R = 8$ when $P = 8$. The $P$ dependence can be explained by the perception-distortion tradeoff [2]: as $P$ increases, $\widehat{\boldsymbol{x}}_{(P)}$ transitions from better perceptual quality to lower $\ell_2$ distortion. PSNR favors $P \to \infty$ (e.g., $\ell_2$ optimality) while the other metrics favor particular combinations of perceptual quality and distortion. The plots in Appendices K.1 and K.2 show zoomed-in versions of $\widehat{\boldsymbol{x}}_{(P)}$ that visually demonstrate the perception-distortion tradeoff at $P \in \{1, 2, 4, 32\}$: smaller $P$ yield sharper images with more variability from the ground truth, while larger $P$ yield smoother reconstructions.

Figure 3 shows zoomed versions of two posterior samples $\widehat{\boldsymbol{x}}_i$, as well as $\widehat{\boldsymbol{x}}_{(P)}$, at $P = 32$ and $R = 8$. The posterior samples show meaningful variations for the proposed method, essentially no variation for Ohayon et al.'s cGAN, and vertical or horizontal reconstruction artifacts for Adler & Öktem's cGAN and the Langevin method, respectively. The $\widehat{\boldsymbol{x}}_{(P)}$ plots show that these artifacts are mostly suppressed by sample averaging with large $P$.

Figure 4 shows examples of $\widehat{\boldsymbol{x}}_{(P)}$, along with the corresponding pixel-wise absolute errors $|\widehat{\boldsymbol{x}}_{(P)} - \boldsymbol{x}|$ and pixel-wise SD images $(\frac{1}{P} \sum_{i=1}^{P} (\widehat{\boldsymbol{x}}_{(P)} - \widehat{\boldsymbol{x}}_i)^2)^{1/2}$, for $P = 32$ and $R = 8$. The absolute-error image for the Langevin technique looks more diffuse than those of the other methods in the brain region. The fact that it is brighter in the air region (i.e., near the edges) is a consequence of minor differences in sensitivity-map estimation. The pixel-wise SD images show a lack of variability for the E2E-VarNet, which does not generate posterior samples, as well as Ohayon et al.'s cGAN, due to mode collapse. The Langevin pixel-wise SD images show localized hot-spots that appear to be reconstruction artifacts.

Appendices K.1 and K.2 show other example MRI recoveries with zoomed pixel-wise SD images at $R = 4$ and $R = 8$, respectively. Notably, Figures K.10 and K.11 show strong hallucinations for Langevin recovery at $R = 8$, as highlighted by the red arrows.

Table 3: Average PSNR, SSIM, LPIPS, and DISTS of $\widehat{\boldsymbol{x}}_{(P)}$ versus $P$ for $R = 8$ MRI

| | PSNR↑ | | | | | | SSIM↑ | | | | | |
|---|---|---|---|---|---|---|---|---|---|---|---|---|
| Model | $P{=}1$ | $P{=}2$ | $P{=}4$ | $P{=}8$ | $P{=}16$ | $P{=}32$ | $P{=}1$ | $P{=}2$ | $P{=}4$ | $P{=}8$ | $P{=}16$ | $P{=}32$ |
| E2E-VarNet (Sriram et al. [37]) | **36.49** | - | - | - | - | - | 0.9220 | - | - | - | - | - |
| Langevin (Jalal et al. [19]) | 32.17 | 32.83 | 33.45 | 33.74 | 33.83 | 33.90 | 0.8725 | 0.8919 | 0.9031 | 0.9091 | 0.9120 | 0.9137 |
| cGAN (Adler & Öktem [8]) | 31.31 | 32.31 | 32.92 | 33.26 | 33.42 | 33.51 | 0.8865 | 0.9045 | 0.9103 | 0.9111 | 0.9102 | 0.9095 |
| cGAN (Ohayon et al. [23]) | 34.89 | 34.90 | 34.90 | 34.90 | 34.91 | 34.92 | 0.9222 | 0.9217 | 0.9213 | 0.9211 | 0.9211 | 0.9210 |
| cGAN (Ours) | 32.32 | 33.67 | 34.53 | 35.01 | 35.27 | 35.42 | 0.9030 | 0.9199 | 0.9252 | **0.9257** | 0.9251 | 0.9246 |

| | LPIPS↓ | | | | | | DISTS↓ | | | | | |
|---|---|---|---|---|---|---|---|---|---|---|---|---|
| Model | $P{=}1$ | $P{=}2$ | $P{=}4$ | $P{=}8$ | $P{=}16$ | $P{=}32$ | $P{=}1$ | $P{=}2$ | $P{=}4$ | $P{=}8$ | $P{=}16$ | $P{=}32$ |
| E2E-VarNet (Sriram et al. [37]) | 0.0575 | - | - | - | - | - | 0.1253 | - | - | - | - | - |
| Langevin (Jalal et al. [19]) | 0.0769 | 0.0619 | 0.0579 | 0.0589 | 0.0611 | 0.0611 | 0.1341 | 0.1136 | 0.1086 | 0.1119 | 0.1175 | 0.1212 |
| cGAN (Adler & Öktem [8]) | 0.0698 | 0.0614 | 0.0623 | 0.0667 | 0.0704 | 0.0727 | 0.1407 | 0.1262 | 0.1252 | 0.1291 | 0.1334 | 0.1361 |
| cGAN (Ohayon et al. [23]) | 0.0532 | 0.0536 | 0.0539 | 0.0540 | 0.0534 | 0.0540 | 0.1128 | 0.1143 | 0.1151 | 0.1155 | 0.1157 | 0.1158 |
| cGAN (Ours) | 0.0418 | **0.0379** | 0.0421 | 0.0476 | 0.0516 | 0.0539 | 0.0906 | **0.0877** | 0.0965 | 0.1063 | 0.1135 | 0.1177 |

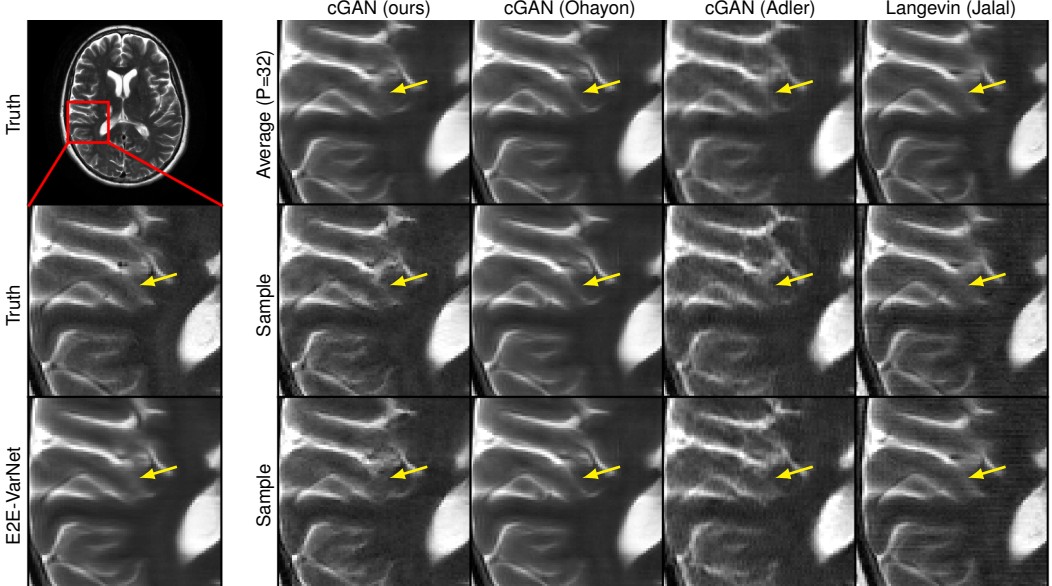

Figure 3: Example $R = 8$ MRI reconstructions. Arrows show meaningful variations across samples.

### 4.3 Inpainting experiments

In this section, our goal is to complete a large missing square in a face image.

**Data.** We used $256 \times 256$ CelebA-HQ face images [29] and a centered $128 \times 128$ missing square. We randomly split the dataset, yielding $27\,000$ training, $2\,000$ validation, and $1\,000$ testing images.

**Architecture.** For our cGAN, we use the CoModGAN generator and discriminator from [9] with our proposed $\mathcal{L}_{1,\mathsf{SD},P_\text{train}}$ regularization. Unlike [9], we do not use MBSD [29] at the discriminator.

**Training/validation/testing.** We use the same general training and testing procedure described in Section 4.2, but with $\beta_\text{adv} = 5\text{e-}3$, $n_\text{batch} = 100$, and 110 epochs of cGAN training. Running PyTorch on a server with 4 Tesla A100 GPUs, each with 82 GB of memory, the cGAN training takes approximately 2 days. FID was evaluated on $1\,000$ test images using $P = 32$ samples per measurement. To avoid the bias that would result from evaluating CFID on only $1\,000$ images (see Appendix J.1), CFID was evaluated on all $30\,000$ images with $P = 1$.

**Competitors.** We compare with the state-of-the-art CoModGAN [9] and Score-based SDE [20] approaches. For CoModGAN, we use the PyTorch implementation from [44]. CoModGAN differs from our cGAN only in its use of MBSD and lack of $\mathcal{L}_{1,\mathsf{SD},P_\text{train}}$ regularization. For Song et al.'s SDE, we use the authors' implementation from [45] with their pretrained weights and the settings they suggested for the $256 \times 256$ CelebA-HQ dataset.

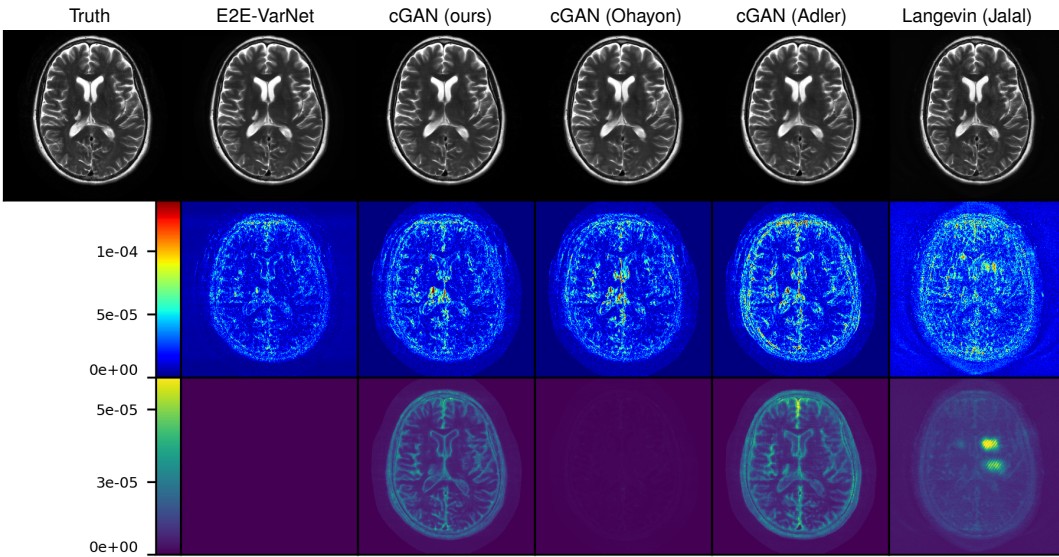

Figure 4: Example $R = 8$ MRI reconstructions with $P = 32$. Row one: $P$-sample average $\widehat{\boldsymbol{x}}_{(P)}$. Row two: pixel-wise absolute error $|\widehat{\boldsymbol{x}}_{(P)} - \boldsymbol{x}|$. Row three: pixel-wise SD $\left(\frac{1}{P}\sum_{i=1}^{P}(\widehat{\boldsymbol{x}}_i - \widehat{\boldsymbol{x}}_{(P)})^2\right)^{1/2}$.

Table 4: Average results for inpainting: FID was computed from $1\,000$ test images with $P = 32$, while CFID was computed from all $30\,000$ images with $P = 1$

| Model | CFID↓ | FID↓ | Time (128)↓ |
|---|---|---|---|
| Score-SDE (Song et al. [20]) | 5.11 | 7.92 | 48 min |
| CoModGAN (Zhao et al. [9]) | 5.29 | 8.50 | **217 ms** |
| cGAN (ours) | **4.69** | **7.45** | **217 ms** |

**Results.** Table 4 shows test CFID, FID, and 128-sample generation time. The table shows that our approach gave the best CFID and FID, and that the cGANs generated samples 13 000 times faster than the score-based method. Figure 5 shows an example of five generated samples for the three methods under test. The samples are all quite good, although a few generated by CoModGAN and the score-based technique have minor artifacts. Some samples generated by our technique show almond-shaped eyes, demonstrating fairness. Additional examples are given in Appendix K.3.

## 5 Conclusion

We propose a novel regularization framework for image-recovery cGANs that consists of supervised-$\ell_1$ loss plus an appropriately weighted standard-deviation reward. For the case of an independent Gaussian posterior, we proved that our regularization yields generated samples that agree with the true-posterior samples in both mean and covariance. We also established limitations for alternatives based on supervised-$\ell_2$ regularization with or without a variance reward. For practical datasets, we proposed a method to auto-tune our standard-deviation reward weight.

Experiments on parallel MRI and large-scale face inpainting showed our proposed method outperforming all cGAN and score-based competitors in CFID, which measures posterior-approximation quality, as well as other metrics like FID, PSNR, SSIM, LPIPS, and DISTS. Furthermore, it generates samples thousands of times faster than Langevin/score-based approaches.

In ongoing work, we are extending our approach so that it can be trained to handle a wide range of recovery tasks, such as MRI with a generic acceleration and sampling mask [46], or inpainting with a generic mask. We are also extending our approach to other applications, such as super-resolution, deblurring, compressive sensing, denoising, and phase retrieval.

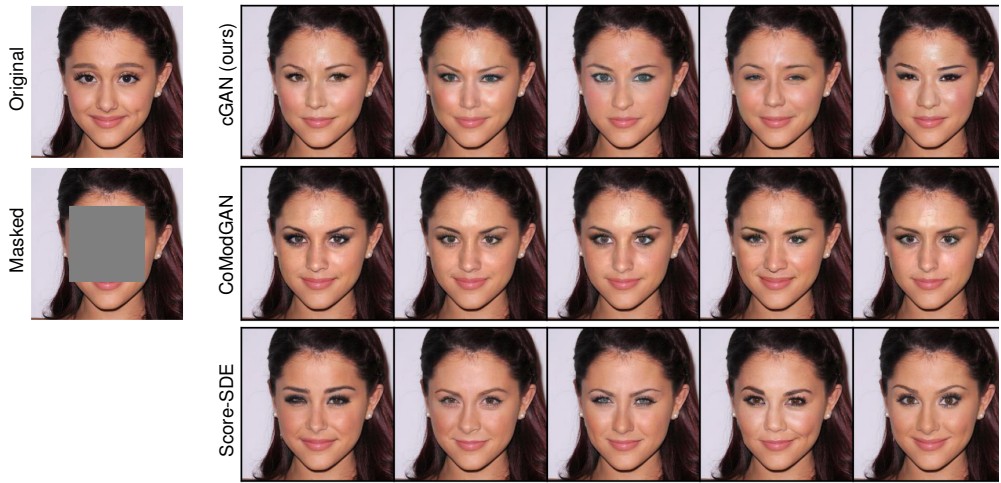

Figure 5: Example of inpainting a $128 \times 128$ square on a $256 \times 256$ resolution CelebA-HQ image.

**Limitations.** We acknowledge several limitations of our work. First, while our current work focuses on how to build a fast and accurate posterior sampler, it's not yet clear how to best exploit the resulting posterior samples in each given application. For example, in MRI, where the posterior distribution has the potential to help assess uncertainty in image recovery, it's still not quite clear how to best convey uncertainty information to radiologists (e.g., they may not gain much from pixel-wise SD images). More work is needed on this front. Second, we acknowledge that, because radiologists are risk-averse, more studies are needed before they will feel comfortable incorporating generative deep-learning-based methods into the clinical workflow. Third, we acknowledge that the visual quality of our $R = 8$ MRI reconstructions falls below clinical standards. Fourth, some caution is needed when interpreting our CFID, FID, and DISTS perceptual metrics because the VGG-16 backbone used to compute them was trained on ImageNet data. Although there is some evidence that the resulting DISTS metric correlates well with radiologists' perceptions [41], there is also evidence that ImageNet-trained features may discard information that is diagnostically relevant in medical imaging [47]. Thus our results need to be validated with a pathology-centric radiologist study before they can be considered relevant to clinical practice.

## Acknowledgments and Disclosure of Funding

The authors are funded in part by the National Institutes of Health under grant R01-EB029957 and the National Science Foundation under grant CCF-1955587.

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
