# OpenReview forum: "A Regularized Conditional GAN for Posterior Sampling in Image Recovery Problems"
_NeurIPS.cc/2023/Conference — NeurIPS 2023 poster_

### Official Review · Reviewer_qA2y · 2023-07-02

**Soundness:** 3 good
**Presentation:** 3 good
**Contribution:** 3 good
**Rating:** 6
**Confidence:** 3

**Summary:**

This paper introduces a regularization term, comprising an L1 penalty and a standard-deviation reward, for the conditional GAN with Wasserstein loss. The objective is to ensure the generation of high-quality samples for a specific input observation y. The proposed approach is evaluated on two image recovery tasks: MRI reconstruction and face image inpainting. The numerical results illustrate its effectiveness in comparison to alternative methods.

**Strengths:**

1) The proposed regularization term is simple and easy to implement and be added to GAN models;
2) Discussed different regularization options and demonstrated why they did not work in this case;
3) It includes theoretical analysis and empirical results;
4) The overall writing and presentation are good and easy to follow.

**Weaknesses:**

1) The novelty of this paper sounds a little bit limited. The supervision-L1 to regularize conditional GAN is proposed by previous work. So it looks like it only added a new reward term on top of previous work.

2) The two experimental tasks are all linear image restoration tasks. However, it is curious how the proposed method would work on nonlinear tasks.

3) It would also be interesting to see whether the proposed method would work on a more complicated real-world dataset for the inpainting task.

4) The current experimental image datasets are low resolution. Will it still outperform other models in terms of quality and computational cost on higher-resolution image datasets?

5) How to determine the hyper-parameter: belta_adv for different datasets?

6) Minor issue: there are some writing inconsistencies (e.g., the capitalization of the headline of 4.2 /4.3 is different from the headline of 3.1/3.2).


#------------------------------------------------------------------------------------------------#

#------------------------------------------------------------------------------------------------#

Most of my previous concerns have been addressed by the authors during the rebuttal, thus I have increased my rating accordingly. However, it would be great if the authors could add some experiments about non-linear image inverse problems.

**Questions:**

See [Weaknesses].

**Limitations:**

Yes, the authors have discussed their limitations in the Conclusion section.

---

> ### Author Rebuttal · Authors · 2023-08-09
>
> - **Weakness**:
> Novelty is a bit limited. Only added a new reward term.
> **Response**:
> Please see the global rebuttal.
>
> - **Weakness**:
> Only linear inverse problems were demonstrated.  How would it perform on nonlinear ones?
> **Response**:
> Our method should work equally well for nonlinear inverse problems, since it makes no assumption about model linearity. In fact, it does not assume any particular forward model, and instead infers the posterior distribution from a supervised training set $\\{(\\boldsymbol{x}_i,\\boldsymbol{y}_i)\\}$. We chose to experiment with MRI and large-scale inpainting because they are well-studied problems with strong baselines.
>
> - **Weakness**:
> How would the proposed method work on a more complicated real-world dataset for the inpainting task?
> **Response**:
> Our primary contribution is a cGAN regularization that prevents mode-collapse. To our knowledge, there is nothing about our regularization that favors particular datasets over others. For good reconstruction quality, a cGAN must have a good generator, but the literature contains many examples of high-performance generators for a wealth of applications.
>
> - **Weakness**:
> The current experimental image datasets are low resolution. Will it still outperform other models in terms of quality and computational cost on higher-resolution image datasets?
> **Response**:
> For MRI, we use 384x384 pixel, 8 coil, complex-valued brain images, which contain 2.4 million unknowns. This size is standard for clinical practice, so it would not be correct to call it low resolution. For inpainting, we use the 256x256 version of CelebA-HQ, which is the standard resolution of CelebA-HQ in the literature. For example, the 256x256 size was needed access a public pretrained CelebA-HQ model for the Score SDE baseline.  In any case, there is no reason to believe that our proposed regularization would work poorly with larger images.
>
> - **Weakness**:
> How to choose $\beta_{\mathsf{adv}}$?
> **Response**:
> In our experience, $\beta_{\mathsf{adv}}$ does not require precise tuning. Our strategy has been to set $\beta_{\mathsf{adv}}$ to roughly equate the size of the adversarial loss term and the regularization term.

---

> > ### Comment · Reviewer_qA2y · 2023-08-21
> >
> > Thanks for taking the time to address my previous questions and concerns. After reading your feedback, the global rebuttal, and reviews from other reviewers and discussions, most of my concerns have been addressed and I do value the quality of this paper; consequently, I have raised my rating.

---

### Official Review · Reviewer_ur5t · 2023-07-03

**Soundness:** 3 good
**Presentation:** 4 excellent
**Contribution:** 3 good
**Rating:** 7
**Confidence:** 3

**Summary:**

This work uses conditional GANs for solving image recovery problems including face-inpainting and accelerated MRI. In particular, a regularization scheme is proposed that tackles the lack-of-diversity issue related to posterior sampling from cGANs. The authors show that the proposed regularization allows sampling from the true posterior under the assumption of a Gaussian signal prior. Empirically the method outperforms a U-Net trained end-to-end, existing cGANs and two score/diffusion/Langevin approaches in terms of accuracy and the latter also significantly in terms of speed.

**Strengths:**

1. The paper is written very clearly and is well organized.
2. The proposed regularization significantly improves over existing cGAN approaches and hence will be of interest for the community.
3. The theoretical discussion of a toy example with Gaussian signal prior is insightful and contributes to the understanding of the proposed regularization.

**Weaknesses:**

1. The authors argue that sampling from the posterior distribution alleviates any concerns about fairness and facilitates uncertainty quantification and further that uncertainty quantification in MRI, and fairness in face-generation, are both of paramount importance. In Appendix A they show that a classifier trained on images from the prior distribution maintains its performance when averaged over many samples from the posterior but not when evaluated on a point estimate.
However, the empirical experiments in Section 4 do not evaluate these points in detail. Regarding uncertainty quantification for MRI Figure 4 shows one example of the pixel-wise standard deviation (SD) but a discussion on the usefulness and practical applicability is missing. To me it seems that in the depicted SD map the model is equally uncertain about every edge in the image making me wonder about the usefulness of this map to a radiologist.
Regarding fairness in face-generation 5 different reconstructions of one image are shown and the authors argue that the fact that some samples show almond-shaped eyes demonstrates fairness.
    Here, a more detailed evaluation would strengthen the contribution although I acknowledge that this might be out the scope for a single paper.

2. In terms of reconstruction accuracy the U-Net is shown to be outperformed but I would assume that a stronger baseline trained end-to-end like a Var-Net https://arxiv.org/abs/1704.00447 would perform significantly better in terms of PSNR and SSIM. Further, the conditioning makes the model lose the advantage of unconditioned generative models for solving inverse problems to be independent from the forward map.
    Since the authors also do not convincingly demonstrate cGANs to be superior in terms of fairness and uncertainty estimation (see previous pont), I wonder about the general usefulness of cGANs as proposed in this work.
    However, I acknowledge that this work significantly outperforms previous cGAN based approaches and thus is a valuable contribution to the field of generative models for solving inverse problems. Also the authors discuss the dependence on a specific forward model as a limitation and refer to ongoing work that removes this constraint.

**Questions:**

No additional questions.

**Limitations:**

Limitations are discussed adequately by the authors.

---

> ### Author Rebuttal · Authors · 2023-08-09
>
> - **Weakness**:
> The experiments did not demonstrate fairness or classifier calibration advantages due to posterior sampling. Also, it's not clear how a radiologist would benefit from a pixel-wise standard-deviation map.
> **Response**:
> Our goal for this paper was to design a fast and accurate posterior sampler and to evaluate the quality of its samples using principled metrics like CFID. In ongoing work, we are currently studying the *application* of posterior sampling to particular fields.  This includes studying how posterior sampling can be leveraged to ensure fairness, how it can be leveraged to quantify the uncertainty of downstream tasks like classification, and how uncertainty information can be best communicated to radiologists. Although these application-related questions are important, we feel that they are outside the scope of our NeurIPS submission.
>
> - **Weakness**:
> The UNet is a weak baseline.  Better to compare to the E2E-VarNet.
> **Response**:
> Please see our global rebuttal.
>
> - **Weakness**:
> Due to a dependence on the forward model and the lack of demonstration of fairness and uncertainty estimation, the general usefulness of cGANs in posterior sampling is unclear.
> **Response**:
> It's true that the dependence of cGANs on the forward model is a disadvantage, although we are addressing this limitation in ongoing work.  But unconditional approaches like Langevin/score/diffusion methods have their own disadvantages, such as much slower sample-generation time and the need to have an explicit model for the likelihood function $p(\\boldsymbol{y}|\\boldsymbol{x})$.

---

> > ### Comment · Reviewer_ur5t · 2023-08-16
> >
> > Thank you for your response. I have read the other reviews and responses.
> > I appreciate the discussion with reviewer N79n and the limitations added to the final version of the paper and further the additional results for the VarNet as a baseline, which shows that the quantitative performance of the approach is competitive with state-of-the-art end-to-end learning.
> > I also acknowledge that the main goal of this work was to propose a fast and accurate posterior sampler and that the application of this sampler for uncertainty quantification is outside the scope of this paper. Hence, I increased the contribution socre from 2 to 3 and the overall score from 6 to 7.

---

### Official Review · Reviewer_N79n · 2023-07-05

**Soundness:** 4 excellent
**Presentation:** 2 fair
**Contribution:** 3 good
**Rating:** 6
**Confidence:** 4

**Summary:**

The paper presents a new method based on conditional GANs (cGANs) for generating posterior samples for solutions of inverse problems. The motivation comes from the fact that standard approaches to inverse problems typically use point estimates, which due to the perception-distortion tradeoff will lead to blurry images. Conversely, reconstruction via sampling can be used to navigate the perception-distortion tradeoff by either sampling small counts to have sharp images, or averaging many samples to approximate the point estimate reconstruction.

All of the above is known from the previous literature. The present paper's specific contributions largely center on the loss function and regularization design for cGANs to avoid posterior collapse and increase sample quality. The regularizer design is based on matching the means and covariances of the samples to the means and covariances of the true distribution. The manuscript chooses to compute these statistics over the $\ell_1$ norm rather than the $\ell_2$ norm, and the paper shows that $\ell_2$ norm can in fact lead to collapse.

In numerical experiments the paper demonstrates superior FID and conditional FID performance for the proposed method, indicating that it is superior at distribution matching than the previous methods considered. For distortion metrics, the experiments demonstrate that superior distortion performance can be achieved by averaging over samples.

**Strengths:**

**Originality**
- Interesting new regularization method for training cGANs for image restoration tasks.

**Quality**
- The paper's quantitative results indicate SotA performance across a variety of tasks for both MRI reconstruction and image inpainting.
- Principled use of VGG weights instead of Inception weights for FID calculation and correlations to radiologist perception.
- A derivation motivating the use of both the supervised standard deviation reward and the application of this award over $\ell_1$ norms.

**Clarity**
- Excellent presentation of perception-distortion tradeoff and its relevance to imaging inverse problems.
- Detailed outline of the proposed method with the appendices, as well as relevant mathematics concerning the various tasks in the results.

**Significance**
- There is substantial interest in the image reconstruction/restoration community now, much of which has focused on conditional diffusion models. The establishment of competitive performance in this space with cGANs (as well as the superior performance to past cGAN methods) will be of interest.

**Weaknesses:**

**Quality**
- The MRI images still demonstrate substantial artifacts from all methods at 8X acceleration. It is possible to find a number of features and fine details that are missing, so I doubt these would be accepted for clinical use.
- Although VGG may correlate better with radiologist scores, it is still an ImageNet feature backbone and contains nullspaces that may be poor for the tasks under consideration (Kynkäänniemi, 2023).
- There is a literature dedicated to so-called "plug-and-play" methods that are relevant to the present work, but was not considered. See (Venkatakrishnan, 2013) and follow-up papers.
- The U-Net is used as a surrogate for point-estimate methods in the MRI example, but it is known in the MRI community that the U-Net is a poor reconstructor. A more standard comparison would be to use the End-to-End Variational Network (Sriram 2020), which is available from the fastMRI GitHub repository. This comparison was included in the cited paper of (Jalal 2021).
- The competing methods for inpainting seem out of date. Dozens of methods based on conditional probabilistic diffusion have emerged over the last year (e.g., DDRM (Kawar 2022) and follow-ups), so I do not find these experiments as strong as the MRI ones.

1. Kynkäänniemi, Tuomas, et al. "The Role of ImageNet Classes in Frechet Inception Distance." ICLR (2023).
2. Venkatakrishnan, Singanallur V., Charles A. Bouman, and Brendt Wohlberg. "Plug-and-play priors for model based reconstruction." 2013 IEEE global conference on signal and information processing. IEEE, 2013.
3. Sriram, Anuroop, et al. "End-to-end variational networks for accelerated MRI reconstruction." Medical Image Computing and Computer Assisted Intervention–MICCAI 2020: 23rd International Conference, Lima, Peru, October 4–8, 2020, Proceedings, Part II 23. Springer International Publishing, 2020.
4. Kawar, Bahjat, et al. "Denoising diffusion restoration models." Advances in Neural Information Processing Systems 35 (2022): 23593-23606.

**Questions:**

I am positive on this work, but a number of aspects on the presentation could be improved. I would appreciate the authors' responses on improving the following, mostly related to the limitations of the work and contextualization with existing literature on medical image reconstruction.

1. Although sampling-based reconstruction can do a better job of distribution-matching and uncertainty estimates, the artifacts introduced are unpredictable. For medical imaging, calibrated radiologists can work around predictable artifacts such as blurring. In fact, even point-estimate reconstruction has not fully taken over the medical field, because simple linear reconstructions have more predictable artifacts than deep learning point estimates. The use of generative sampling compounds this effect and is likely to be substantial limitation on the present work. I think a discussion on this should be added as limitation (some sketch of the proposed discussion would be appreciated during the rebuttal).
2. The paper primarily examines distribution matching over ImageNet features, which are likely to be quite different than the features useful for medical imaging. (Kynkäänniemi, 2023) shows the presence of perceptual nullspaces that could very well ignore small pathologies. Even the self-supervised solution of (Kynkäänniemi, 2023) is likely to be of benefit for medical imaging analysis, because self-supervised features rely on large objects at the center of the image.
3. Although the classifier matching proof in Appendix A seems theoretically sound, it relies on a few limits and calibration assumptions that are likely to deviate from reality, so I am skeptical this is a grounded proof that classifiers trained on fully-sampled images will perform adequately on the sampled images.
4. The paper could benefit with a comparison to the End-to-End variational network, which is a standard comparison method in the MRI literature and was previously used in the Jalal paper. Although the present manuscript has shown superiority to (Jalal, 2021) on metrics, sampling-based methods have not yet taken the general field of medical image reconstruction, so the comparison is still valid.
5. A cursory overview of the Plug-and-Play reconstruction literature would provide a more complete picture of previous work.

**Limitations:**

As discussed under Questions, the paper could expand its discussion of limitations, including the issues of 1) issues artifact consistency posterior sampling methods with human observers and 2) the use of metrics based on ImageNet feature alignment for medical imaging tasks.

---

> ### Author Rebuttal · Authors · 2023-08-09
>
> - **Weakness**:
> At 8x acceleration, it is possible to find a number of features and fine details that are missing, so I doubt these would be accepted for clinical use.
> **Response**:
> 8x acceleration is perhaps too aggressive for clinical practice, but keep in mind that we are zooming the plots significantly. For example, the E2E-VarNet paper (Sriram'20) shows only non-zoomed 8x plots that look good from casual inspection.  But by looking closely, one can spot artifacts and missing details that would be clearly evident in a zoomed plot.
>
> - **Weakness**:
> The competing methods for inpainting seem out of date. Dozens of diffusion methods have emerged over the last year (e.g., DDRM and follow-ons).
> **Response**:
> Please see the global response.
>
> - **Question**:
> Radiologists are okay with blurring, but not with unpredictable artifacts. Will radiologists benefit from the use of generative sampling?
> **Response**:
> There exists a fundamental uncertainty in what one can infer about the true image from sub-sampled k-space measurements, and the posterior distribution gives a full description of that uncertainty. But we agree that it's not yet clear how to best distill that uncertainty information for radiologists, and this limits the short-term impact of our work. Ignoring uncertainty altogether, however, does not seem to be a good solution. Thus we are optimistic that radiologists will eventually be persuaded to integrate uncertainty information into their workflows. In ongoing work, we are investigating various ways of doing so, including rigorous confidence bounds on classifier outputs, counterfactual generation, and visualizations of dominant uncertainty eigenspaces.
>
> - **Question**:
> Are metrics computed using a VGG backbone appropriate for MRI?
> **Response**:
> This is an excellent question. (Kastryulin'22) studied 35 contemporary MRI metrics and found that, of the four methods that best correlated with radiologists perceptions, two used features from ImageNet-trained VGG. But we agree that improved metrics are likely result from non-ImageNet backbones, and we hope to soon see studies that validate them with radiologists. In the meanwhile, however, we worry that readers may not trust our evaluations if they are based on novel metrics. In any case, if our final paper is accepted, we will discuss the limitations of using ImageNet-trained backbones for MRI performance metrics.
>
> - **Question**:
> The classifier matching proof relies on a perfectly calibrated classifier.  Can this be generalized to practical classifiers?
> **Response**:
> Based on our ongoing work, the answer seems to be "yes". For our Neurips submission, though, we feel that this is a bit out of scope.
>
> - **Question**:
> The UNet is a weak baseline. Compare to an end-to-end (E2E) VarNet, as in (Jalal'21)?
> **Response**:
> Please see the global rebuttal.
>
> - **Question**:
> Discuss plug-and-play methods?
> **Response**:
> Since our paper focuses on posterior-sampling, we did not review point-estimation approaches like compressed sensing, PnP/RED, unrolled networks, deep equilibrium, etc.  That said, adding noise to each PnP/RED iteration gives a posterior sampler known as the unadjusted Langevin algorithm (Durmus'17), which is a precursor to the Langevin/score/diffusion techniques that we consider.  We will discuss these PnP connections in the final paper if accepted.

---

> > ### Comment · Reviewer_N79n · 2023-08-14
> >
> > Thank you for your response. I have also read the other reviews and responses.
> >
> > However, I still think there is an outstanding issue with the limitations. Although Sriram '20 may have had issues at 8X acceleration, we also know that 8X acceleration is not clinically acceptable for the E2E VarNet, even though it is acceptable for 4X (Recht et al., 2020, Muckley et al., 2021). My point was that the present work still does not seem to resolve the 8X usability issue.
> >
> > The response on the questions of radiologist reading through blurring does not directly address my concern. In general, radiology is extremely risk-averse. It is much easier to just scan a little bit longer than to use to a risky method, and medical practitioners will generally opt for the less-risky method. Uncertainty maps can help, but it is easier to just not deal with the problem.
> >
> > As for the ImageNet feature alignment, this reference, like others in the field, primarily relies on radiologist preference. It can be difficult to evaluate observer performance. For clinical deployment, the critical component is not what images radiologists like, but whether they can see the pathology. Radiologist rankings of aspects such as SNR or CNR often don't correlate to whether they can see pathology - for example, abdominal imaging in general is very bad for SNR, but still shows many pathologies.
> >
> > So, in summary, I still think the limitations should be expanded for both points I made above, to be more conservative in the claims that the improved results probably need to be validated with a pathology-centric radiologist study, and with the current experiments don't assess that performance. If the limitations discussion were expanded to incorporate both of these, I could improve my score.
> >
> > 1. Recht, Michael P., et al. "Using deep learning to accelerate knee MRI at 3 T: results of an interchangeability study." AJR. American journal of roentgenology 215.6 (2020): 1421.
> > 2. Muckley, Matthew J., et al. "Results of the 2020 fastMRI challenge for machine learning MR image reconstruction." IEEE transactions on medical imaging 40.9 (2021): 2306-2317.

---

> > > ### Author Response · Authors · 2023-08-15
> > >
> > > We thank the reviewer for the continued discussion and opportunity to improve our submission.
> > >
> > > - **Reviewer**:
> > > Although Sriram '20 may have had issues at 8X acceleration, we also know that 8X acceleration is not clinically acceptable for the E2E VarNet, even though it is acceptable for 4X (Recht et al., 2020, Muckley et al., 2021). My point was that the present work still does not seem to resolve the 8X usability issue.
> > > **Response**:
> > > We chose to focus on 8X because it yields more uncertainty and thus more interesting images than 4X. But we agree that 8X acceleration may be outside of the clinical acceptability window for our method and perhaps all existing methods. Experimentally, we have observed that our performance gains over our competitors are similar at 4X compared to 8X. However, our submission included only a 4X version of Table 1 (that includes CFID, FID, APSD, and run-time) in Appendix J.1. Thus, if our paper is accepted, we will do the following. *First, we will add a limitation statement acknowledging that at 8X acceleration, the recoveries generated by our method likely fall below clinical quality. Second, we will add a complete reporting of 4X results to our paper's supplementary materials. In particular, we will add a 4X version of Table 2 (that includes PSNR, SSIM, LPIPS, and DISTS) as well as 4X versions of Figure 1, Figure 2, and Figure K.1 through K.6.*
> > >
> > > - **Reviewer**:
> > > The response on the questions of radiologist reading through blurring does not directly address my concern. In general, radiology is extremely risk-averse. It is much easier to just scan a little bit longer than to use to a risky method, and medical practitioners will generally opt for the less-risky method. Uncertainty maps can help, but it is easier to just not deal with the problem.
> > > **Response**:
> > > Although it's possible to lower the acceleration rate in certain applications (e.g., 2D brain and MSK imaging), it is not possible in others, such as breath-held cardiac cine or dynamic applications in general. For this reason, we believe that it is important to quantify the fundamental uncertainties associated with MR image recovery, and this is the goal of posterior sampling.  In any case, we agree that radiology is extremely risk-averse, and so *we will add a limitation statement acknowledging that more rigorous studies are needed before radiologists feel comfortable adopting generative deep-learning-based methods in clinical practice.*
> > >
> > > - **Reviewer**:
> > > As for the ImageNet feature alignment, this reference, like others in the field, primarily relies on radiologist preference. It can be difficult to evaluate observer performance. For clinical deployment, the critical component is not what images radiologists like, but whether they can see the pathology. Radiologist rankings of aspects such as SNR or CNR often don't correlate to whether they can see pathology - for example, abdominal imaging in general is very bad for SNR, but still shows many pathologies. So, in summary, I still think the limitations should be expanded for both points I made above, to be more conservative in the claims that the improved results probably need to be validated with a pathology-centric radiologist study, and with the current experiments don't assess that performance. If the limitations discussion were expanded to incorporate both of these, I could improve my score.
> > > **Response**:
> > > We agree that the critical component of image quality is whether radiologists can see the pathology. Thus, *we will add a limitation statement acknowledging that our results, as well as the metrics on which we measured performance, need to be validated with a pathology-centric radiologist study before they can be considered relevant to clinical practice.*
> > >
> > > Please let us know if our proposed remedies fall short.  We thank you again for your thorough review, continued discussion, and help in improving our submission.

---

> > > > ### Comment · Reviewer_N79n · 2023-08-15
> > > >
> > > > I think the proposed limitations satisfy my primary concerns, and I have increased my score to 6.

---

### Official Review · Reviewer_pyUh · 2023-07-13

**Soundness:** 2 fair
**Presentation:** 3 good
**Contribution:** 3 good
**Rating:** 5
**Confidence:** 3

**Summary:**

This paper proposed a cGAN (conditional generative adversarial network) method for general image inverse problems (IR). Conventional cGAN methods for IR usually suffer from mode collapse that the generators learned to ignore latent inputs and are unable to generate diverse image reconstruction samples. Motivated by existing work, this work proposes to incorporate multiple latent input to the GAN model during training to mitigate the mode collapse effect. Accordingly, a sophisticated designed supervised training loss function is also presented in this work. The numerical results show that the proposed method generates comparable or even better results to cGAN and score-based baseline methods on compressive MRI and image inpainting.

**Strengths:**

1, The paper is overall well motivated, clearly written and easy to follow.

2, The proposed posterior sampling-based method is effective for generating multiple samples compared to some score-based methods. The multiple sampling can also be applied for model uncertainty estimation, which is important for many medical imaging problems.

3, The theoretical interpretation on the mean and covariance of predicted posterior distribution is informative, though it is sometimes unrealistic for most real-world imaging problems.


**Weaknesses:**

1, One contribution of this paper seems still to be based on the idea of Adler & Oktem [9], where multiple latent vectors are as inputs to the discriminator and generator module instead of only one input. In the meantime, as pointed out by the author as well, adding extra regularization term in the adversarial GAN loss during training is not completely new.

2, This proposal ignores a body of work for posterior sampling based on Bayesian deep learning such as Monte Carlo dropout, deep ensembles, etc. There’s also a lack of numerical comparisons with these methods, making the baseline methods not sufficient enough.

3, While some ablation studies provided in its original submission, the effects of regularization parameter $\beta_{SD}$ and number $P$ on the final imaging reconstruction are still not comprehensive and could be improved. For example, a visual illustration or numerical comparison of several proposed sampling results due to different settings of $\beta_{SD}$ is preferable.

4, The assumption for proposition 3.1 that samples $\widehat x_i$ and $x$ are independent Gaussian conditioned on y is not true for imaging applications, weekend the theoretical contribution.

5, Some technical details and notations are confusing and could be improved in their current state.


**Questions:**

1), There seems to have a gap between proposition 3.3 and the gradient descent on the $\beta_{SD}$ and could be clarified for reader’s better understanding. For example, there is no clear formular between $\beta_{SD}$ and the ratio in Eq 20., making the derivative of Eq. 21-23 a bit confusing. It is unclear how Eq. 21-22 is formulated and how this led to the update of $\beta_{SD}$. Please clarify this part in the revision.

2), At the same line above, it is unclear why the number of latent $P$ is not the same for training and testing (e.g., line 166,” for some $P_{\text{val}} $ that does not need to match $P_{\text{train}} $.”). Is this means the performance of this proposal for each image is sensitive to the choice of value $P$? It is unclear when there is a lack of enough validation dataset to finetune $\beta$ or if there is a mismatch between train and testing dataset. As a result, the effects of hyper-parameter settings for different settings are unclear.

3), The paper lacks sufficient discussion on the CS-MRI experiments.  For instance, based on the description of Sec. 4.2, “we transformed $x_t$ to the Fourier domain”, seems the authors do not follow the trends of other methods in the fast MRI leaderboard that directly perform down-sampling on the kspace data. This seems like an important distinction and must be made clear in the paper. There is no discussion on how the measurement noise is modeled or how it affects the reconstruction performance. Also, while there are some technical discussions on the CSMRI formulation in the appendix, there is no description of how the forward model simulation in this paper differs from the actual forward model of the real MRI system.

4), One benefit of posterior sampling is for model uncertainty estimation. While this work can generate multiple samples and show the pixel-wise standard deviation in Figure 4, it is difficult to evaluate if it can indeed reflect the mean-absolute error for image reconstruction. It would be great to have a quantitative result for this study.

5), There is a lack of comparison to more recent diffusion model-based IR methods, where many have been demonstrated good posterior sampling performance for various imaging tasks, such as [R1, R2, R3].

6),  The run time comparison to  the selected Langevin dynamic baselines seems to be suboptimal, as there are many efficient sampling strategies can be applied for IR diffusion/score-based methods, where fewer iteration steps could result in appealing reconstruction [R3].

7), It is unclear which forward model/dataset are used for Figure 2 in the paper. If it is plotted using images, it is unclear how is the curve in theory plotted. Moreover, it would be more informative if Figure2 can be directly represented by $\epsilon_1/\epsilon_P$ and $\beta_{SD}$.

[R1] Score-based diffusion models for accelerated MRI, Chung. etal, 2021.

[R2] Diffusion posterior sampling for general noisy inverse problems, Chung. etal, 2022.

[R3] Pseudoinverse-Guided Diffusion Models for Inverse Problems, Song. etal, 2023.


**Limitations:**

The authors discussed their work's limitations in the submission.

---

> ### Author Rebuttal · Authors · 2023-08-09
>
> - **Weakness**:
> Adding extra regularization to the GAN loss is not completely new.
> **Response**:
> Please see the global rebuttal.
>
> - **Weakness**:
> No discussion of Monte-Carlo dropout or deep ensembles.
> **Response**:
> Posterior samplers aim to accurately sample from the true posterior $p(\boldsymbol{x}|\boldsymbol{y})$, which is a property of the data $\\{(\boldsymbol{x}_i,\boldsymbol{y}_i)\\}$ and not any trained network. In contrast, Monte-Carlo dropout and deep ensembles heuristically perturb a given deep network and quantify its uncertainty according to the variance induced by those perturbations; see (Gal'16).  They make no attempt to accurately sample from the posterior, which is a non-trivial task with high dimensional data such as images.  Image posterior sampling is tackled by cGANs, cNFs, cVAEs, and Langevin/scores/diffusion methods. To be consistent with that literature, we didn't discuss/compare to Monte-Carlo dropout or deep ensembles. But if the reviewer has particular papers in mind, we could include a discussion/comparison in the final version (if accepted).
>
> - **Weakness**:
> The effect of $\\beta_{\\mathsf{SD}}$ and $P$ on final reconstruction are not well described.
> **Response**:
> Neither are tunable parameters.  There is a correct value for $\\beta_{\\mathsf{SD}}$, and our method estimates it as described in Section 3.2.  For more on $\\beta_{\\mathsf{SD}}$ and $P$, please see our responses below.
>
> - **Question**:
> What is the relationship between Prop. 3.3 and the update on $\beta_{\mathsf{SD}}$?
> **Response**:
> Just after equation (20), we write that the ratio "$\\mathcal{E}\_1 / \\mathcal{E}\_P$ grows with the SD reward weight $\\beta_{\\mathsf{SD}}$,"
> and we provide evidence of this behavior in Figure 2. Thus, if the ratio $\\mathcal{E}\_1/\\mathcal{E}\_P$ is smaller than the value $2P/(P+1)$ specified in Prop. 3, then $\\beta_{\\mathsf{SD}}$ is too low. Conversely, if the ratio is larger than $2P/(P+1)$, then $\\beta_{\\mathsf{SD}}$ is too high. The $\\beta_{\\mathsf{SD}}$ update (23) simply increases or decreases $\\beta_{\\mathsf{SD}}$ depending on whether $\\mathcal{E}_1/\\mathcal{E}_P$ is smaller or larger than $2P/(P+1)$.
>
> - **Question**:
> Why does $P$ differ between training and testing?
> **Response**:
> $P$ is simply the number of generated samples. Changing $P$ does not affect the quality of those samples. When training, Prop. 3.1 shows that any $P\geq 2$ can be used. In practice, larger values of $P$ consume more GPU memory, and so we use the minimum value of 2. When testing, the choice of $P$ depends on the task. If the goal is to generate a single posterior sample, then $P=1$ suffices. If the goal is to approximate the conditional mean (i.e., MMSE) estimate, then larger $P$ will give lower MSE (see Prop. 3.3) but consume more computational resources. When validating during modeling training, we roughly estimate the MSE at each epoch, for which we used $P=8$ to strike a balance between accuracy and computation.
>
> - **Question**:
> Are you using a non-standard MRI setup?
> **Response**:
> No, we are using a standard MRI setup. Because the fastMRI samples vary in dimension, we center-crop them to a consistent dimension of 384x384. The cropping must be done in the image domain, not the k-space, else the sampling resolution will change. This is common practice: see page 15 of the fastMRI paper https://arxiv.org/pdf/1811.08839.pdf. In summary, we start with fully sampled k-space data, convert to the image domain, center-crop it to 384x384, convert back to the k-space, and then sub-sample it. No additional measurement noise is added.
>
> - **Question**:
> Does pixel-wise standard deviation reflect the mean-absolute error (MAE) of image reconstruction?
> **Response**:
> Not quite. Say $x$ is a single pixel of the unknown image. The posterior distribution $p(x|\boldsymbol{y})$ is the probability distribution of $x$ given the measurements $\boldsymbol{y}$. Then for a given pixel estimate $\hat{x}$, we have MSE($\hat{x}$) $=\int |x-\hat{x}|^2 \\, p(x|\boldsymbol{y}) dx$ and MAE($\hat{x}$) $=\int |x-\hat{x}| \\, p(x|\boldsymbol{y}) dx$.  The posterior variance, $\int |x-\mathrm{E}\\{x|\boldsymbol{y}\\}|^2 \, p(x|\boldsymbol{y}) dx$, equals the minimum possible MSE for any $\hat{x}$, and the standard deviation is simply its square root.  Note that MSE and MAE are functions of $\hat{x}$ while the posterior standard deviation is not.
>
> - **Question**:
> Why not compare to more recent diffusion models like (Chung'21), (Chung'22), or (Song'23)?
> **Response**:
> Please see the global response.
>
> - **Question**:
> Why didn't you use sampling strategies that make diffusion/score-based methods more efficient?
> **Response**:
> We did not want to modify the existing Langevin and Score-SDE techniques from (Jalal'21) and (Song'21) because we could be blamed if the resulting modified techniques under-performed our approach. So we used the authors' recommendations. But we note that more recent diffusion methods like DDRM at 20 steps are still 275x slower than our cGAN (see global rebuttal).
>
> - **Question**:
> Which dataset was used for Figure 2? And can the figure be represented by $\\mathcal{E}\_1 / \mathcal{E}\_P$ and $\beta_\text{SD}$?
> **Response**:
> For Figure 2, we used the fastMRI brain data at $R=4$. We will describe this in the final paper if accepted.  We plot $1/\mathcal{E}\_P$ rather than $\mathcal{E}\_1 / \mathcal{E}\_P$, because the former contains information about $\mathcal{E}\_1$ that is lost in the latter.  But in the dB domain, the two curves are related by a simple vertical shift.  Finally, Figure 2 prints the value of $\beta_\text{SD}$ in the title of each subplot.

---

> > ### Comment · Reviewer_pyUh · 2023-08-21
> > **Official Comment by Reviewer pyUh**
> >
> > The reviewer thank you the authors for addressing the  reviewers’ concerns. After going through the results in the new adding materials and the discussions between other reviewers, I tend to accept this submission. However, in the context of Medical imaging and specific to MRI it’s unclear whether perceptual metrics are really relevant. It is clear that E2E-VarNet still works the best in terms of PSNR by 1.07 dB. In addition, seemingly the choice of $P$ may affect the final reconstruction results a lot, making the practical usage of this algorithm unclear. As a result, I can not increase my score as its current state. That is being said the review does not feel disappointed if this paper dose not got accepted eventually.

---

> > > ### Author Response · Authors · 2023-08-21
> > >
> > > Thanks for your comments and continued discussion.
> > >
> > > **"In the context of Medical imaging and specific to MRI it's unclear whether perceptual metrics are really relevant"**
> > >
> > > - What matters in MRI is the ability of the radiologist to make the correct diagnosis.  Studies like [A,B,C,D] have shown that well-chosen perceptual metrics correlate far better to radiologists perceptions than traditional metrics like PSNR.
> > >
> > > **"It is clear that E2E-VarNet still works the best in terms of PSNR by 1.07 dB"**
> > >
> > > - First, please see our previous comment about how PSNR is not a meaningful visual quality metric for MRI.
> > > - Second, the E2E-VarNet generates only a point estimate, not posterior samples.  The subject of our paper is posterior sampling, not point estimation.  Posterior sampling has many applications, only one of which is constructing point estimates.
> > > - Third, when used to construct point estimates, our approach beats the E2E-VarNet in *every* tested metric except PSNR.
> > >
> > > **"The choice of $P$ may affect the final reconstruction results a lot, making the practical usage of this algorithm unclear"**
> > >
> > > - Our paper is about fast and accurate posterior sampling, and $P$ is simply the number of samples that the user chooses to generate.
> > > - The effect of $P$ on reconstruction quality only manifests when creating point estimates, and the main subject of our paper is not point estimation.
> > > - Even if we do focus on point estimation, then the effect of $P$ is relatively clear.  It is a direct way to navigate the perception-distortion tradeoff, which is a fundamental tradeoff in image reconstruction.  Given a chosen image-quality metric, experiments like those in Table 2 clearly show how $P$ is best chosen.
> > >
> > >
> > > [A] S. Kastryulin, J. Zakirov, N. Pezzotti, and D.V. Dylov, “Image quality assessment for magnetic resonance imaging,” arXiv:2203.07809, 2022.
> > >
> > > [B] M.S. Treder, R. Codrai, and K.A. Tsvetanov, "Quality assessment of anatomical MRI images from generative adversarial networks: Human assessment and image quality metrics," Journal of Neuroscience Methods, vol. 374, 2022.
> > >
> > > [C] Mason, A., et al., 2020. Comparison of Objective Image Quality Metrics to Expert Radiologists’ Scoring of Diagnostic Quality of MR Images. IEEE TMI 39, 1064–1072
> > >
> > > [D] H. Liu, W. Zhou, and R. Narayan. "Perceptual quality assessment of medical images." In Encyclopedia of Biomedical Engineering, pp. 588-596. 2019.

---

### Author Rebuttal · Authors · 2023-08-09

We sincerely thank the reviewers for the time spent reading and reviewing our manuscript.  The feedback is very useful and will help to improve the quality of the final paper if it is accepted.  In this global rebuttal we respond to questions that were raised by multiple reviewers.

- **Weakness**:
Novelty is a bit limited. Supervised-L1 regularization has been proposed and so you are only adding a reward term.
**Response**:
Yes, using supervised-L1 regularization in cGANs is well known. Our contributions include proving that the typical L1 and L2 regularizations actually *induce* mode collapse in cGANs, and proposing a novel regularization that prevents mode collapse. Our framework also addresses the tuning of the standard-deviation reward weight, which is a difficult problem in practice.  Our regularization provably leads to true-posterior sampling in simple cases and empirically yields high-fidelity samples in real-world imaging applications.

- **Question**:
Isn't the UNet a weak baseline?
**Response**:
There are many variations of the UNet. The one we used for both our baseline point-estimator and for our generator is considerably more powerful than the UNet described in the fastMRI paper (Zbontar'18), to which comparisons are typically made. First, our UNet operates on complex multicoil images and we coil-combine its output, while the UNet in (Zbontar'18) operates on single-coil magnitude images after coil-combining its input. Second, we employ several architectural modifications, as detailed in Appendix I.1.2 of our paper. Third, we apply data-consistency as described in Appendix H. In any case, our main motivation to compare to this UNet is that it differs from our proposed method only in the training loss, and therefore illustrates the effect of our proposed loss/regularization.

- **Question**:
Why not compare to an end-to-end (E2E) trained VarNet, as in (Jalal'21)?
**Response**:
Thanks, this is a good idea.  We trained an E2E VarNet using the authors' code at 8x acceleration with the same sampling mask and training data used for the other methods under test. The results in the attached pdf show our method outperforming the E2E VarNet in all metrics except PSNR. We will include these results in the final paper. Like (Jalal'21), we used ESPIRIT-based ground truth for training and performance evaluation, as justified in Appendix A.1 of (Jalal'21).

- **Question**:
Why not compare to more recent diffusion models like (Chung'21), DPS (Chung'22), DDRM (Kawar'22), or PiDGM (Song'23)?
**Response**:
We chose well-known diffusion baselines rather than the absolute latest methods. Regarding the cited papers:
-- The MRI method from (Chung'21) is not practical in that it requires the inputs to be precisely normalized, which the authors do by peeking at the ground truth. See line 52 in their code:
https://github.com/HJ-harry/score-MRI/blob/main/inference_multi-coil_hybrid.py. We found that their method fails when the normalization constant is estimated from sub-sampled data.
-- As for DPS, the main contribution is a way of handling significantly *noisy* inverse problems, whereas our MRI and inpainting problems have very high SNR. Also, it is not clear how to best apply this method to complex-valued multicoil MRI, which is a nontrivial undertaking.
-- We attempted to run DDRM as suggested for our inpainting problem, but no pre-trained CelebA-HQ model is currently available (see https://github.com/bahjat-kawar/ddrm/issues/24).
-- For PiDGM, we could not find public code.
In any case, a persistent issue with diffusion methods is their slow evaluation speed.  For example, DDRM at 20 steps would be the fastest among the above methods, but after running it on our hardware with batch size of 128 and image size of 256x256, we found that it is still 275x slower than our cGAN.

- **General comment**:
Our main contribution is a novel cGAN regularization for posterior sampling. Our regularization can be used with any generator architecture, the choice of which will strongly affect performance in any given application.  Although the performance we achieved on MRI and inpainting tasks was good, it should not be taken as a ceiling, because our choice of generator architecture can certainly be improved upon. For example, the Langevin/score/diffusion models and E2E-VarNet effectively use a series of many CNNs, with data-fidelity steps in-between, whereas we use only a single CNN.

---

### Decision · Program_Chairs · 2023-09-21

**Decision:**

Accept (poster)

**Comment:**

The paper presented a conditional GAN method for general imaging problems that addresses the posterior collapse problem using an L1 instead of L2 penalty. The four reviewers all agreed the paper should be accepted, with three reviewers raising their scores in light of the author discussion period. The reviewers recognized the impressive performance of the regularization technique for sampling images and one reviewer appreciated the added discussion of areas where the method may have limitations.